# CauFed-CLIP: Causal Federated Vision-Language Models for Domain Generalization

## Abstract

Although visual language models (VLMs) have achieved remarkable success, applying them directly in federated learning (FL) faces key challenges: high communication/computation costs and poor generalization due to client data heterogeneity. To tackle these, we propose **CauFed-CLIP**, a novel Causal-based Federated Contrastive Language-Image Pre-training model. Our model reduces overhead by freezing the VLM backbone and training a lightweight causal module on clients. To enhance generalization, our model employs a progressive causal mechanism. It first disentangles observed features ($x$) into domain-invariant ($s$) and domain-variant ($z$) representations, aided by global and local guidance to suppress their spurious correlations. From this disentangled foundation, it then infers the underlying causal "concept" ($c$)—a quasi-invariant latent variable that represents the essence of a category and holds a weak causal link with the domain ($z$). Ultimately, relying solely on this pure concept '$c$' for prediction allows the model to transcend superficial statistics and grasp the core causal logic. Experiments on six benchmarks across natural and medical domains show that CauFed-CLIP consistently outperforms state-of-the-art FL methods, especially in cross-domain generalization.

## 1 Introduction

In recent years, the success of Vision Language Models (VLMs) is driven by large-scale pretraining on massive datasets (Xu et al., 2024). However, rising data privacy regulations increasingly challenge this centralized training paradigm (Bakare[1] et al., 2024). FL emerges as a promising solution, enabling collaborative model training without sharing raw data (McMahan et al., 2017). Nevertheless, applying large-scale VLMs within the resource-constrained setting of FL creates a direct conflict, presenting two critical bottlenecks (Kuang et al., 2024; Wu et al., 2025): 1) the computational bottleneck: Client devices (e.g., smartphones) often lack the capacity for full VLMs training. 2) the communication bottleneck: Exchanging the massive VLM parameters in each round consumes prohibitive bandwidth, drastically slowing down training.

Collectively, these challenges severely hinder the deployment of VLMs in real-world federated settings (Guo et al., 2023b; Saha et al., 2025a). To overcome these bottlenecks, a prevailing strategy is to freeze the large VLM backbone as a universal feature extractor, while clients only train and exchange lightweight task modules (Lu et al., 2023; Wu et al., 2025). Although this efficiency gain makes the deployment of VLMs in FL feasible, a more fundamental challenge emerges: significant domain shifts across clients (Chen et al., 2024; Bai et al., 2024). These discrepancies, originating from diverse domains (e.g., photos, sketches, cartoons), cause even powerful VLMs to learn spurious, domain-specific correlations from the features they extract (Varma et al., 2024a; Ye et al., 2024; Ma et al., 2025). The direct consequence is a severe degradation in generalization ability, causing the model to become "paranoid" and biased towards the domains seen during training. When such a model is deployed on unseen clients, its performance plummets, often rendering it unusable. Existing studies on domain generalization typically focus on improving model architectures (e.g., attention mechanisms) (Saha et al., 2025b; Wu et al., 2025; Ma et al., 2025) or introducing complex regularizations (Li et al., 2020; Zhang et al., 2024a). Other approaches attempt to construct centralized proxy datasets, a strategy that fundamentally violates FL's core principle of privacy preservation (Kalra et al., 2023; Wang et al., 2023; Liu et al., 2024). We argue that these methods either fail to address the root of the problem or do so at the cost of privacy. To truly overcome this fundamen-

tal generalization crisis, models must transition from "**pattern recognition**" (learning "*what*") to "**causal reasoning**" (understanding "*why*"). Recent work has attempted to obtain invariant features by erasing spurious ones through causal intervention (Li et al., 2025) or prompt fine-tuning (Gong et al., 2024; Ma et al., 2025). However, their causal assumptions have limitations—they tend to treat invariant and domain-specific features as independent, thus potentially overlooking the subtle causal connections between them.

To this end, we innovatively introduce causal inference to FL and propose the **CauFed-CLIP** model. Initially, a lightweight module disentangles observed features ($x$) into a domain-invariant representation ($s$) and a domain-variant representation ($z$). However, we find that the simple assumption that only the invariant representation determines the label (i.e., $s \rightarrow y$) is one-sided. For example, a real-world "cow" differs from a cartoon "cow" in its core features because the "cartoon" domain itself carries an "abstraction" causal effect. Conversely, when an animal is erased from a desert background, people are more inclined to guess it was a "camel" than a "cow," indicating a subtle link between the domain and the concept. Inspired by these observations, we propose a more refined causal structure: we hypothesize the existence of a pure "concept" ($c$) that is core-stable yet adaptively fine-tuned by the domain. This concept serves as the common root cause driving the invariant representation ($s$), the variant representation ($z$), and the final label ($y$). More critically, we argue that a weak causal relationship exists between this concept ($c$) and the domain environment ($z$). By modeling and performing inference on this causal graph, our model can distill truly robust, essential features and focus on learning domain-general knowledge, thereby achieving excellent domain generalization capabilities on unseen clients. The main contributions are:

1. We propose the CauFed-CLIP model, which addresses the two key challenges of efficiency and generalization in federated VLMs within a unified framework. By adopting a parameter-efficient strategy, the framework overcomes efficiency bottlenecks, and through the introduction of a fine-grained causal model, it guides the learning of underlying causal structures—fundamentally enhancing the model's domain generalization capability.

2. We design an innovative prompt-guided mechanism that drives causal disentanglement through efficient and interpretable semantic supervision. This mechanism leverages a shared global prompt and privacy-preserving local prompts to create semantic "anchors," providing strong signals for learning causal representations.

3. We conducted comprehensive experiments on six domain generalization datasets. The results show that CauFed-CLIP significantly outperforms state-of-the-art (SOTA) methods in terms of accuracy, communication efficiency, and domain generalization capability.

## 2 RELATED WORK

### 2.1 FEDERATED LEARNING

FL addresses data silo and privacy issues via distributed collaborative training (McMahan et al., 2017), yet struggles with client drift due to data heterogeneity (Xiao et al., 2024). Common solutions include proximal regularization (Li et al., 2020), local batch normalization (Li et al., 2021b), and contrastive or graph-based representation alignment (Li et al., 2021a; Xiao et al., 2024). However, these often fall short under significant domain or covariate shifts.

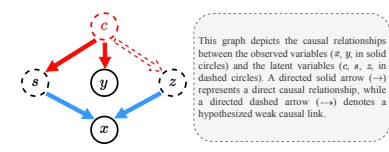

Figure 1: The proposed causal graph.

### 2.2 CLIP IN FL: FROM EFFICIENT FINE-TUNING TO DOMAIN GENERALIZATION

Recent efforts integrate VLMs like CLIP into FL through parameter-efficient fine-tuning (PEFT) to reduce computational costs, by learning only lightweight prompts or adapters locally (Pan et al., 2024; Saha et al., 2025a; Chen et al., 2024; Guo et al., 2023a; Yang et al., 2023; Wu et al., 2025; Shi et al., 2024). While improving efficiency, PEFT methods tend to learn spurious domain-specific features, harming generalization (Varma et al., 2024b). Some works pursue invariant features via

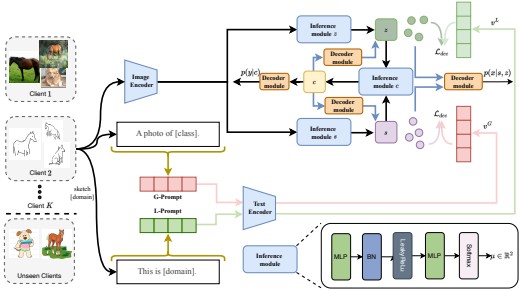

Figure 2: Causal-based Federated Contrastive Language-Image Pre-training (CauFed-CLIP) model.

test-time prompting (Ma et al., 2025) or causal learning (Chen et al., 2023; Zhang et al., 2025), but often assume feature independence, overlooking their underlying causal relationships.

In contrast, our work is the first to introduce a more complete causal inference framework to federated VLMs. We advocate for modeling the deep causal structure behind the features, positing a core hypothesis: a pure, domain-invariant "concept" acts as the common root cause for the observed invariant features, variant features, and the final label. By performing inference on this causal graph (Fig. 1), our model aims to transition from learning "superficial correlations" to understanding "deep causality," thereby fundamentally enhancing its domain generalization. For a more comprehensive review of related work, see Appendix F.

## 3 METHOD

### 3.1 PROBLEM SETTING

We consider an FL setting with $K$ clients $\{C_1, \ldots, C_K\}$, each holding a local private dataset $\mathcal{D}_k = \{(x_{k,j}, y_{k,j})\}_{j=1}^{n_k}$. The core challenge is data heterogeneity, where local data distributions differ ($P(\mathcal{D}_{k'}) \neq P(\mathcal{D}_k)$ $for$ $k' \neq k$). Each client's dataset $\mathcal{D}_k$ is partitioned into training ($\mathcal{D}_k^{\text{train}}$), validation ($\mathcal{D}_k^{\text{val}}$), and test ($\mathcal{D}_k^{\text{test}}$) sets. The traditional FL objective is to learn a global model $f_\theta(\cdot)$ that minimizes the average loss over the training clients' test data:

$$\min_\theta \frac{1}{K} \sum_{k=1}^{K} \mathcal{L}_k(f_\theta; \mathcal{D}_k^{\text{test}}), \tag{1}$$

where $\mathcal{L}_k(f_\theta; \mathcal{D}_k^{\text{test}}) = \frac{1}{|\mathcal{D}_k^{\text{test}}|} \sum_{(x,y) \in \mathcal{D}_k^{\text{test}}} \ell_k(f_\theta(x), y)$ denotes the average loss on the test set of client $k$, and $\ell_k$ is the client-specific loss function.

However, real-world deployment requires generalization to unseen clients, leading to the problem of **federated domain generalization**. Our goal is to learn a model $f_\theta$ that not only performs well on the training clients but also generalizes effectively to $M$ unseen clients, $\{C_{K+1}, \ldots, C_{K+M}\}$. The data $\mathcal{D}_m$ ($m \in \{K+1, \ldots, K+M\}$) from unseen clients may exhibit significant distribution shifts, such as **domain shift** (e.g., paintings vs. photos) or **covariate shift** (e.g., sunny vs. snowy car images). Our goal is to minimize the model's expected loss on unseen clients:

$$\min_\theta \frac{1}{M} \sum_{m=K+1}^{K+M} \mathcal{L}_m(f_\theta; \mathcal{D}_m). \tag{2}$$

### 3.2 FRAMEWORK OVERVIEW

#### 3.2.1 CAUSAL FORMALISM.

To tackle FL's data heterogeneity and domain shift, we build a structural causal model (SCM) (Fig. 1), modeling causal relationships between latent variables $V := \{c, s, z\}$ and observable variables $(x, y)$ for robust cross-domain predictions. We define each variable in the graph as follows:

- Observed Data ($x$): Raw image features extracted from the VLM on the client side.

- Task Label ($y$): The true label of the image.

- Domain-Variant Representation ($z$): A latent variable that captures domain-specific features (e.g., lighting, background, style) and is the root cause of domain shift.

- Invariant Representation ($s$): A latent variable that represents the essential semantic features of an object, independent of specific domains.

- Concept ($c$): A quasi-invariant latent variable that represents the essence of a category and possesses domain-adaptivity.

Based on these definitions, we establish the following causal relationships (i.e., the directed edges in the graph):

1. $c \to y$, $c \to s$: The pure invariant concept $c$ is the common root cause of both the label $y$ and the invariant feature $s$. This establishes our core objective: to learn the causal predictor $p(y|c)$, which is inherently stable and invariant across all domains.

2. $c \dashrightarrow z$: We hypothesize a weak causal link from the pure concept $c$ to the domain-variant representation $z$. This connection explains why the same concept (e.g., a "cow") manifests differently across domains (e.g., "real photo" vs. "cartoon"), acknowledging that concept and domain are not entirely independent.

3. $s \to x$, $z \to x$: The observed feature $x$ is generated by both the invariant feature $s$ and the domain-variant feature $z$. As a collider node, $x$ thus entangles task-relevant information ($s$) with domain-specific noise ($z$), making the direct prediction $p(y|x)$ unreliable and prone to poor generalization.

### 3.2.2 IDENTIFIABILITY ANALYSIS.

After constructing the causal graph, a central theoretical question arises: *Can our target causal relationship $p(y|c)$ be uniquely identified from the observed data $(x, y)$?* Since $c$, $s$, and $z$ are latent variables that cannot be directly observed, the identifiability argument hinges on the following core assumptions:

1. **Structural Correctness of the Causal Graph:** We assume that the proposed causal graph (as shown in Fig. 1) is correct, meaning it fully captures all causal relationships among the variables $x, y, c, s$, and $z$, and that there are no unmeasured confounders.

2. **Parametric Model Sufficiency:** We assume our framework's neural networks (Fig. 2) can sufficiently approximate true data distributions like $p(x|s, z, c)$ and $p(y|c)$.

3. **Causal Faithfulness:** We assume that conditional independencies in the data arise solely from the graph's structure, not from coincidental cancellations of causal pathways.

Under these assumptions, the identifiability of our target, $p(y|c)$, hinges on addressing a key back-door path between the observed features $x$ and the label $y$: $y \leftarrow c \dashrightarrow z \to x$. This path is "open" because it introduces spurious, non-causal associations between $x$ and $y$ through the common cause $c$ and the mediator $z$. For example, this may lead the model to incorrectly associate domain-specific features (such as the "cartoon style" represented by $z$) with the class label (e.g., "animal" represented by $y$), which is the core of the domain shift problem.

According to Pearl's Back-door Criterion, in order to block such confounding and identify the true causal effect, we must condition on a set of variables that blocks all back-door paths (Pearl, 1995). In our model, this critical back-door path can be blocked by conditioning on the latent domain-relevant representation $z$.

Therefore, $p(y|c)$ is theoretically identifiable, provided that our learning algorithm can effectively infer and disentangle the latent variable $z$ from the observed data $x$. This analysis provides the theoretical foundation for our method, mandating that its success depends on this precise disentanglement capability.

### 3.2.3 CAUSALLY-INFORMED REPRESENTATION LEARNING VIA PROMPT-GUIDED VARIATIONAL INFERENCE.

Guided by our causal identifiability analysis, we design a learning framework to infer the latent variables $c$, $s$, and $z$ from the observed data $x$. As shown in Fig. 2, our framework innovatively integrates the principles of variational inference (VI) with the power of VLM, supervised by a novel prompt-guiding mechanism. CauFed-CLIP comprises the following key components:

1. CLIP Backbone: To ensure efficiency in the federated setting, we leverage a pretrained CLIP model as a frozen feature extractor. Clients only train and communicate our lightweight causal module, drastically reducing both computational and communication overhead.

2. Inference Network $q_\phi$: This network disentangles the input image features x to infer the posterior distribution of the latent variables, $q_\phi(c, s, z|x)$.

3. Prompt-Guiding Mechanism: We design a shared **Global Prompt (G-Prompt)** and a private **Local Prompt (L-Prompt)** to generate semantic anchor vectors, $v^G$ and $v^L$, which capture domain-invariant and domain-specific semantics, respectively. As shown in Eq. (3), a symmetric contrastive loss, ($\mathcal{L}_{\text{prompt}}$), then enforces a strong alignment between these latent representations and their corresponding anchors ($s$ towards $v^G$ and $z$ towards $v^L$). This targeted semantic supervision provides a powerful and direct signal to drive the causal disentanglement.

$$\hat{y} = [0, 1, \ldots, B-1], \ \mathbf{I} = \text{sim}(z, v^L), \ \mathbf{T} = \text{sim}(v^L, z), \ \mathcal{L}_{\text{prompt}} = \frac{1}{2}\left(\ell(\mathbf{I}, \hat{y}) + \ell(\mathbf{T}, \hat{y})\right), \quad (3)$$

where $B$ is the number of samples in a batch, $\text{sim}(\cdot, \cdot)$ denotes the scaled cosine similarity matrix, and $\ell$ is the cross-entropy loss.

**Variational Inference and Objective Function.** Based on the causal identifiability analysis, our goal is to learn a model that can uncover the latent variables $c, s, z$, and ultimately enable robust causal prediction $p(y|c)$. Theoretically, this can be achieved by maximizing the log-likelihood of the observed data $(x, y)$. However, this likelihood requires integrating over all unknown latent variables, i.e.,

$$\log p(x, y) = \log \int p(x, y, c, s, z) \, dc \, ds \, dz \quad (4)$$

Directly optimizing this objective is intractable due to the complexity of the latent space. To address this, we adopt the method of VI. The core idea of VI is to introduce a parameterized and tractable variational distribution $q_\phi(c, s, z|x, y)$ to approximate the true but intractable posterior $p(c, s, z|x, y)$. We then jointly optimize the generative model (parameterized by $\theta$) and the inference network (parameterized by $\phi$) by maximizing the *Evidence Lower Bound* (ELBO). The ELBO can be written as:

$$\mathcal{L}(\theta, \phi; x, y) = \mathbb{E}_{q_\phi(c, s, z|x, y)}\left[\log \frac{p_\theta(x, y, c, s, z)}{q_\phi(c, s, z|x)}\right] \quad (5)$$

A standard form of the ELBO is constructed based on the inference network $q_\phi(c, s, z|x, y)$. However, when predicting on new samples, the label $y$ is unknown, and therefore the inference network cannot take $y$ as input. Instead, we must use an inference network that conditions only on $x$, i.e., $q_\phi(c, s, z|x)$. To integrate this $x$-only inference network into the ELBO framework, we need to transform the objective function accordingly. By applying Bayes' rule:$q(c, s, z|x, y) = \frac{q(y|x,c,s,z)q(c,s,z|x)}{q(y|x)}$ and incorporating our causal assumption—that $y$ is determined solely by $c$—we derive an equivalent, optimizable objective function. Appendix C provides the full derivation.

$$\mathcal{L} = \mathbb{E}_{p^*(x,y)}\left[\underbrace{\log q_\phi(y|x)}_{(a)} + \frac{1}{q_\phi(y|x)}\underbrace{\mathbb{E}_{q_\phi(c,s,z|x)}\left[p_\theta(y|c)\log\frac{p_\theta(c,s,z)p_\theta(x|s,z)}{q_\phi(c,s,z|x)}\right]}_{(b)}\right] \quad (6)$$

Here, term (a), $\log q_\phi(y|x)$, is a standard supervised prediction loss. Term (b) acts as a causal consistency regularizer, ensuring the prediction from (a) aligns with our predefined causal generative process, rather than relying on simple pattern matching. The core expectation within term (b) measures the alignment between our inference network $q_\phi$ and the generative model $p_\theta$. Given our structured inference network, $q_\phi(c, s, z|x) = q_\phi(s|x)q_\phi(z|x)q_\phi(c|s, z)$, this term can be expanded as follows:

$$\mathbb{E}_{q_\phi(c,s,z|x)} \left[ \log \frac{p_\theta(c)p_\theta(s,z|c)p_\theta(x|s,z)}{q_\phi(s|x)q_\phi(z|x)q_\phi(c|s,z)} \right] = \mathbb{E}_{q_\phi(s|x)q_\phi(z|x)} \left[ \log p_\theta(x|s,z) \right]$$
$$- KL(q_\phi(c|s,z)\|p_\theta(c)) - KL(q_\phi(s|x)\|p_\theta(s|c)) - KL(q_\phi(z|x)\|p_\theta(z|c)) \tag{7}$$

In essence, this decomposition reveals two key components: a reconstruction term $p_\theta(x|s,z)$ that ensures fidelity to the input data, and KL divergence regularizers that promote a structured, disentangled latent space. Thus, our full objective (Eq. 6) compels the model to marry accurate prediction with adherence to the proposed causal structure.

## 4 EXPERIMENTS

This section presents our experimental setup and results. We first describe the datasets and baselines, then show main results with comparisons to SOTA methods, demonstrating our model's superiority. Finally, ablation studies validate the contribution of each component in CauFed-CLIP. Further details and code are in Appendix D.

### 4.1 DATASETS

We evaluate the model's performance in **domain generalization** and **robustness to covariate shift** across multiple challenging benchmarks. For domain generalization, we employ four classic datasets: PACS, which includes four distinct domains—Photo, Art Painting, Cartoon, and Sketch—with 7 categories; OfficeHome, covering four domains—Art, Clipart, Product, and Real-World—with 65 object classes; ModernOffice-31, an extended version of Office-31, comprising four domains—Amazon, Webcam, DSLR, and Synthetic—with 31 categories; and Brain Tumor (BT), a public medical imaging dataset containing four types of tumors, where different imaging devices or patient cohorts can be treated as distinct domains. For covariate shift evaluation, we use CIFAR-10 and CIFAR-100 as in-distribution (IN) training and evaluation data, and further employ CIFAR-10-C and CIFAR-100-C as out-of-distribution test sets. These two datasets are generated by applying 19 types of algorithmic corruptions to the original test sets to simulate real-world covariate shifts. More details regarding the datasets and their splits can be found in Appendix C.

Table 1: Leave-Two-Domain-Out generalization accuracy (%) on PACS, reported as mean $\pm$ standard deviation over 3 independent runs on data partitioned by a Dirichlet distribution (source domain standard deviation is detailed in Appendix D).

| Method | Source | | | | | | | | | | Target | | Avg |
|---|---|---|---|---|---|---|---|---|---|---|---|---|---|
| | S | | | | | A | | | | | | | |
| | $C_1$ | $C_2$ | $C_3$ | $C_4$ | $C_5$ | $C_6$ | $C_7$ | $C_8$ | $C_9$ | $C_{10}$ | C | P | |
| FedCLIP | 85.24 | 88.89 | 86.98 | 82.7 | 76.28 | 90.0 | 79.49 | 83.33 | 97.54 | 100.0 | $94.99_{\pm0.5}$ | $97.46_{\pm0.7}$ | 88.57 |
| FedProx | 92.46 | 86.67 | 90.31 | 87.62 | 91.79 | 89.46 | 83.67 | 87.31 | 84.62 | 88.79 | $91.46_{\pm1.2}$ | $92.19_{\pm1.4}$ | 88.86 |
| FedAVG | 90.93 | 89.45 | 92.92 | 87.36 | 93.08 | 87.93 | 86.45 | 89.92 | 84.36 | 90.08 | $91.56_{\pm1.0}$ | $92.96_{\pm1.2}$ | 89.75 |
| MOON | 91.69 | 83.89 | 93.44 | 88.39 | 92.44 | 88.69 | 80.89 | 90.44 | 85.39 | 89.44 | $91.20_{\pm1.6}$ | $92.46_{\pm0.1}$ | 89.03 |
| FAACLIP | 92.89 | 88.89 | 89.06 | 84.73 | 88.55 | 90.0 | 76.92 | 90.36 | 97.32 | 95.83 | $96.33_{\pm0.3}$ | $98.76_{\pm0.3}$ | 90.82 |
| Ours | 93.13 | 88.89 | 94.79 | 87.28 | 86.54 | 93.33 | 94.87 | 90.91 | 97.32 | 97.74 | $\mathbf{97.63}_{\pm0.3}$ | $\mathbf{99.70}_{\pm0.0}$ | **93.51** |
| | P | | | | | S | | | | | | | |
| | $C_1$ | $C_2$ | $C_3$ | $C_4$ | $C_5$ | $C_6$ | $C_7$ | $C_8$ | $C_9$ | $C_{10}$ | C | A | |
| FedCLIP | 100.0 | 96.97 | 97.96 | 97.06 | 100.0 | 83.46 | 86.11 | 85.94 | 80.91 | 75.64 | $94.09_{\pm0.4}$ | $95.9_{\pm0.3}$ | 91.17 |
| FedProx | 98.68 | 96.88 | 93.64 | 93.04 | 97.91 | 85.55 | 87.06 | 85.64 | 85.04 | 86.93 | $83.28_{\pm0.8}$ | $85.50_{\pm0.6}$ | 89.93 |
| FedAVG | 98.68 | 96.88 | 93.64 | 92.06 | 95.00 | 84.10 | 86.93 | 84.96 | 84.06 | 87.03 | $80.14_{\pm1.0}$ | $82.76_{\pm0.3}$ | 88.86 |
| MOON | 100.0 | 96.88 | 93.64 | 93.04 | 95.00 | 87.00 | 86.33 | 85.64 | 85.04 | 87.00 | $82.21_{\pm0.2}$ | $85.01_{\pm0.4}$ | 89.73 |
| FAACLIP | 100.0 | 96.97 | 96.32 | 98.02 | 100.0 | 82.44 | 88.89 | 85.94 | 84.48 | 79.49 | $96.47_{\pm0.2}$ | $94.86_{\pm0.1}$ | 91.99 |
| Ours | 100.0 | 100.0 | 97.96 | 97.06 | 100.0 | 93.38 | 86.11 | 94.27 | 86.26 | 83.33 | $\mathbf{97.65}_{\pm0.3}$ | $\mathbf{96.37}_{\pm0.5}$ | **94.37** |

### 4.2 BASELINE METHODS

To validate the effectiveness of CauFed-CLIP in handling unseen domain and covariate shift data, we compared it against a series of SOTA FL models. The selected baselines include the classic FedAvg (McMahan et al., 2017); FedProx (Li et al., 2020) and MOON (Li et al., 2021a), which address data heterogeneity; and FedLN (Wei et al., 2022) and FedIIR (Guo et al., 2023b), which target specific

Table 2: Accuracy(%) on multiple unseen domains of the OfficeHome dataset. Bold means the best.

| Method | Source | | Target | | Avg | Source | | Target | | Avg |
|---|---|---|---|---|---|---|---|---|---|---|
| | A | C | R | P | | A | R | C | P | |
| FedCLIP | 75.62 | 77.06 | 87.79 | 88.56 | 82.26 | 75.21 | 88.28 | 65.50 | 88.80 | 79.45 |
| FedProx | 64.88 | 72.94 | 70.25 | 66.66 | 68.68 | 71.07 | 80.00 | 56.66 | 72.94 | 70.17 |
| FedAVG | 67.36 | 71.33 | 71.38 | 67.04 | 69.28 | 67.36 | 82.07 | 57.73 | 71.91 | 69.77 |
| MOON | 67.36 | 71.33 | 71.43 | 67.02 | 69.28 | 73.55 | 86.90 | 63.39 | 86.46 | 77.57 |
| FAA-CLIP | 75.21 | 77.06 | 87.54 | 88.06 | 81.97 | 74.79 | **88.51** | 65.64 | 88.51 | 79.36 |
| Ours | **77.27** | **80.28** | **87.87** | **89.49** | **83.73** | **76.01** | **88.51** | **67.11** | **89.22** | **80.21** |
| | C | R | A | P | Avg | R | P | A | C | Avg |
| FedCLIP | 76.83 | **89.43** | 77.30 | 88.76 | 83.08 | **88.97** | 91.65 | 77.67 | 66.27 | 81.14 |
| FedProx | 72.48 | 78.16 | 66.54 | 72.47 | 72.41 | 75.63 | 80.59 | 61.48 | 54.62 | 68.08 |
| FedAVG | 72.25 | 79.77 | 65.64 | 70.92 | 72.14 | 75.40 | 81.49 | 62.01 | 55.58 | 68.62 |
| MOON | 76.83 | 79.77 | 60.86 | 71.82 | 72.32 | 86.90 | 84.88 | 74.82 | 63.39 | 77.50 |
| FAA-CLIP | 76.38 | **89.43** | 77.54 | 88.51 | 82.97 | 88.18 | 91.20 | 77.50 | 66.21 | 80.77 |
| Ours | **78.90** | 88.74 | **77.96** | **89.24** | **83.71** | **88.97** | **92.10** | **78.17** | **67.56** | **81.90** |

Table 3: Accuracy(%) in the OfficeHome dataset.

| Method | Source | | | Target | Avg | Source | | | Target | Avg |
|---|---|---|---|---|---|---|---|---|---|---|
| | C | P | R | A | | A | P | R | C | |
| FedCLIP | 68.61 | 87.37 | 88.06 | 78.00 | 80.51 | 78.97 | 87.60 | 87.60 | 63.69 | 79.46 |
| FedProx | 64.38 | 79.14 | 78.76 | 65.60 | 71.97 | 73.81 | 80.38 | 80.48 | 57.64 | 73.08 |
| FedAVG | 64.38 | 79.14 | 78.76 | 65.60 | 71.97 | 73.81 | 80.38 | 80.48 | 57.64 | 73.08 |
| MOON | 69.87 | 78.02 | 79.56 | 65.60 | 73.26 | 70.93 | 80.38 | 80.83 | 52.94 | 71.27 |
| PromptFL | **82.75** | 92.13 | 87.73 | 71.88 | 83.62 | 78.75 | 92.71 | 87.27 | 66.66 | 81.35 |
| FAA-CLIP | 76.98 | 90.87 | 88.86 | 78.20 | 83.73 | 81.44 | 91.65 | 90.58 | 66.39 | 82.51 |
| Ours | 76.30 | **92.15** | **90.11** | **78.83** | **84.35** | 76.39 | 90.67 | 89.54 | **67.38** | 81.00 |
| | A | C | R | P | Avg | A | C | P | R | Avg |
| FedCLIP | 78.56 | 68.50 | 87.37 | 87.52 | 80.49 | 78.35 | 68.38 | 87.94 | 87.79 | 80.61 |
| FedProx | 69.07 | 66.21 | 77.79 | 71.64 | 71.18 | 70.93 | 68.73 | 77.73 | 75.42 | 73.20 |
| FedAVG | 69.07 | 66.21 | 77.79 | 71.64 | 71.18 | 70.93 | 68.73 | 77.73 | 75.42 | 73.20 |
| MOON | 71.55 | 67.70 | 79.33 | 71.21 | 72.45 | 67.42 | 69.30 | 76.32 | 75.81 | 72.21 |
| PromptFL | 77.71 | **80.79** | 88.31 | 85.19 | 83.00 | 77.50 | 83.22 | 90.97 | 85.48 | 84.29 |
| FAA-CLIP | **81.03** | 75.49 | 90.81 | 89.34 | 84.29 | 78.76 | 76.98 | 90.64 | 88.11 | 83.62 |
| Ours | 77.69 | 79.13 | **90.91** | **89.54** | **84.32** | 76.64 | 77.06 | 91.20 | **88.47** | 83.34 |

distribution shifts. Furthermore, we included recent methods that also leverage pre-trained models like CLIP, such as FedCLIP (Lu et al., 2023), PromptFL (Guo et al., 2023a), FOOGD (Liao et al., 2024), and FAA-CLIP (Wu et al., 2025).

## 4.3 EXPERIMENTAL RESULTS

### 4.3.1 PERFORMANCE EVALUATION UNDER DOMAIN SHIFT.

This part focuses on exploring the generalization capability of CauFed-CLIP when handling data from different domains. We further divide the evaluation into two scenarios.

**Multiple domains as unseen clients. Results for PACS.** Table 1 presents a performance comparison of our method against several baselines using leave-two-domain-out cross-validation. The experimental setup utilizes a Dirichlet distribution to simulate data heterogeneity among clients. The results show that our method comprehensively outperforms all competing methods in terms of overall performance. Specifically, our method achieves leading average accuracies of 93.51% and 94.37% in the two cross-validation scenarios. It also demonstrates superior generalization across all unseen target domains, with the highest scores on each (e.g., 97.63%, 99.70%, 97.65%, and 96.37%).

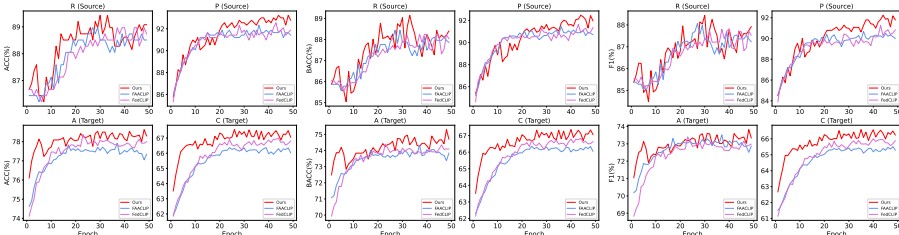

Figure 3: Test accuracy (ACC), balanced accuracy (BACC), and macro-F1 for each communication round on the OfficeHome, with Real World (R) and Product (P) as source domains and Art (A) and Clipart (C) as target domains.

**Results for OfficeHome.** The "leave-two-domains-out" cross-validation results in Table 2 strongly confirm the overall superiority of our method. Across all domain split scenarios, our model not only outperforms others on the unseen target domains, but also achieves the best average accuracy across all domains. For example, in the first group of experiments, our average accuracy of 83.73% significantly surpasses both FedAvg (69.28%) and FAA-CLIP (81.97%), demonstrating robust generalization under complex domain shifts. Fig. 3 further illustrates this advantage. In the generalization task from source domains (R, P) to target domains (A, C), our method (red curve) consistently and significantly outperforms baselines across all key metrics (ACC, BACC, and F1) on the target domains, while also maintaining highly competitive performance on the source domains. This indicates that our model achieves strong generalization without sacrificing local performance.

Table 4: Robustness of methods on CIFAR-100-C ($\alpha = 0.1$).

| Corruption | FedAvg | FedLN | FedIIR | FedCLIP | FOOGD | FAACLIP | Ours |
|------------|--------|-------|--------|---------|-------|---------|------|
| None | 51.67 | 52.48 | 51.63 | 52.87 | 53.84 | 51.09 | **62.55** |
| Brightness | 46.85 | 48.15 | 47.88 | 50.45 | 51.69 | 49.39 | **58.77** |
| Fog | 36.15 | 37.11 | 36.80 | 39.48 | 40.98 | 43.48 | **50.09** |
| glass blur | 20.96 | 27.32 | 19.67 | 37.92 | 27.44 | 33.75 | **35.38** |
| Motion blur | 32.95 | 35.09 | 33.34 | 38.07 | 39.68 | 32.33 | **45.08** |
| Snow | 35.09 | 38.60 | 35.69 | 39.40 | 40.64 | 36.56 | **47.97** |
| Contrast | 26.39 | 27.10 | 26.94 | 31.92 | 30.98 | 30.83 | **39.77** |
| Frost | 32.53 | 35.38 | 33.33 | 37.71 | 38.54 | 37.94 | **46.10** |
| Impulse noise | 22.99 | 24.26 | 21.84 | 29.16 | 26.24 | 28.13 | **35.37** |
| Pixelate | 34.41 | 36.11 | 33.31 | 40.76 | 42.52 | 37.08 | **46.34** |
| Defocus blur | 39.17 | 41.05 | 39.92 | 43.79 | 46.23 | 44.90 | **52.29** |
| Compression | 41.17 | 43.36 | 41.90 | 46.78 | 45.81 | 45.61 | **53.71** |
| Transform | 38.65 | 41.49 | 39.36 | 38.56 | 47.47 | 43.39 | **50.70** |
| Gaussian noise | 21.21 | 24.83 | 21.79 | 32.59 | 28.28 | 30.20 | **35.26** |
| Shot noise | 26.37 | 30.28 | 27.03 | 37.68 | 32.81 | 35.69 | **41.06** |
| Zoom blur | 33.82 | 36.51 | 34.75 | 37.28 | 41.62 | 36.96 | **47.69** |
| Spatter | 42.41 | 43.90 | 42.04 | 42.17 | 49.59 | 38.92 | **54.03** |
| Gaussian blur | 34.18 | 36.29 | 35.39 | 43.31 | 40.63 | 43.57 | **48.23** |
| Saturate | 38.59 | 39.43 | 38.92 | 39.90 | 44.87 | 36.69 | **51.35** |
| Speckle | 26.43 | 30.53 | 27.47 | 36.55 | 32.86 | 35.52 | **41.73** |
| **Avg** | 34.10 | 36.46 | 34.45 | 39.82 | 40.14 | 38.60 | **47.17** |

**Single domain as unseen client.** We employ a Leave-One-Domain-Out strategy on the challenging Brain Tumor (Fig. 4) and OfficeHome (Table 3) datasets. In this setting, where each domain sequentially serves as an unseen test client, our method consistently outperforms all leading baselines in both AUC and accuracy, demonstrating superior generalization to novel domains.

**Performance Evaluation under Covariate Shift.** We evaluate the model's robustness to covariate shift on CIFAR-100-C. As shown in Table 4, our method achieves the best clean accuracy (62.55%) and consistently outperforms baselines across all corruptions,reaching an average accuracy of 47.17%. This demonstrates strong out-of-distribution generalization. Further experiments on domain and covariate shifts are detailed in **Appendix D**.

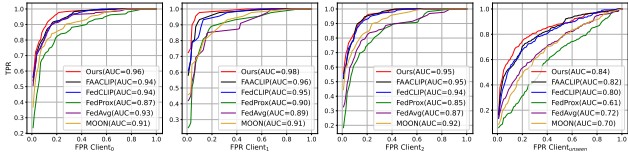

Figure 4: Comparison of ROC curves for different methods on the Brain Tumor (BT) dataset.

## 4.4 ABLATION STUDIES

**Analysis of backbone.** We performed an ablation study on the VLM backbone, testing various architectures. The results in Fig. 5 reveal that performance scales with the backbone's strength. This is likely because a more powerful feature extractor can provide higher-quality initial representations, containing more distinct invariant and variant features with less noise. This, in turn, provides a better foundation for our causal module to perform effective disentanglement and generalization.

**Impact of $K$.** Fig. 6 compares the BACC of different algorithms on unseen domains with varying numbers of clients ($K$=6, 10, and 20). As shown, our model achieves the best results under all conditions, significantly outperforming the others based on the average of 3 independent runs.

**Analysis of Performance under Data Heterogeneity and Covariate Shift.** Fig. 7 shows the results of our ablation study on the CIFAR-10. We simulated varying degrees of data heterogeneity by adjust-

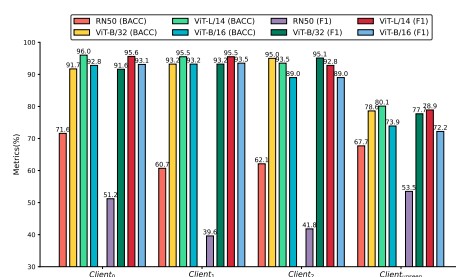

Figure 5: Ablation study on the choice of VLM backbone. Performance comparison (BACC and F1-score) on seen clients ($Client_0$ to $Client_2$) and the unseen client ($Client_{unseen}$) using different Vision Transformer (ViT) and ResNet (RN) models as the frozen feature extractor.

ing the parameter $\alpha$ and introduced brightness corruption (right) to simulate a covariate shift, which was compared against a standard scenario (left). The results indicate that our method demonstrates a consistent and significant advantage under all tested conditions. This finding proves that our model possesses excellent robustness and performance stability when facing the dual challenges of data heterogeneity and covariate shift. We provide additional ablation studies in **Appendix E**.

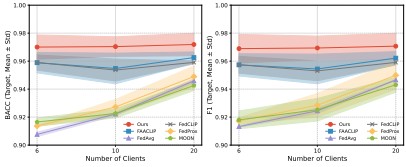
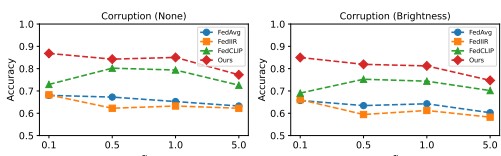

Figure 6: Performance of various baselines under different numbers of clients ($K$).

Figure 7: Accuracy of various methods under covariate shift scenario with different $\alpha$.

## 5 CONCLUSION

In this paper, we proposed CauFed-CLIP, a novel FL framework designed to systematically address the dual challenges of efficiency and domain generalization for Vision-Language Models. By integrating a parameter-efficient strategy with a sophisticated causal disentanglement mechanism, our model successfully learns robust, domain-invariant representations. Extensive experiments demonstrated that CauFed-CLIP significantly outperforms existing SOTA methods, establishing a new and promising direction for building reliable and scalable FL systems.

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

## A   GENERAL STATEMENTS AND BROADER CONTEXT

### A.1   REPRODUCIBILITY STATEMENT

To ensure the reproducibility of our work, we have included all necessary materials and code in the appendix and supplementary materials. Complete theoretical derivations and proofs can be found in Appendix B. The source code for our experiments is provided in the supplementary materials. For further details regarding the experimental setup and hyperparameter configurations, please refer to Appendix D and the documentation in the supplementary materials.

### A.2   DECLARATION ON THE USE OF AI-ASSISTED TECHNOLOGIES

During the writing of this paper, we utilized the Gemini writing assistant to enhance the readability and linguistic accuracy of the text. The tool was employed solely for grammatical improvements and sentence restructuring to ensure clarity of expression. In accordance with the ICLR ethical guidelines, all AI-generated suggestions were rigorously reviewed and edited by the authors, who bear full responsibility for the scientific integrity and entire content of this publication.

## B   DERIVATION OF LEARNING OBJECTIVES

The objective of our causal model 'p' in the federated learning setting is to accurately capture the true global data distribution $p^*(x, y)$. This is achieved by maximizing the log-likelihood of the data, $\mathbb{E}_{p^*(x,y)}[\log p(x, y)]$.

However, the marginal log-likelihood $\log p(x, y)$ is intractable to compute directly, as it requires integrating over all latent variables. To address this, we employ variational inference and instead maximize the Evidence Lower Bound (ELBO), $\mathcal{L}_{p,q}$, which is a lower bound on the log-likelihood derived using Jensen's inequality:

$$
\begin{aligned}
\log p(x, y) &= \log \mathbb{E}_{p(c,s,z)} \left[ p(x, y | c, s, z) \right] \\
&= \log \mathbb{E}_{q(c,s,z|x,y)} \left[ \frac{p(c, s, z, x, y)}{q(c, s, z | x, y)} \right] \\
&\geq \mathbb{E}_{q(c,s,z|x,y)} \left[ \log \frac{p(c, s, z, x, y)}{q(c, s, z | x, y)} \right] \triangleq \mathcal{L}_{p,q}(x, y)
\end{aligned}
\tag{8}
$$

Our goal is thus to maximize the expectation of this ELBO with respect to the true data distribution: $\mathbb{E}_{p^*(x,y)}[\mathcal{L}_{p,q}(x, y)]$.

To make this objective more practical and to explicitly isolate the predictive term for classification, we rewrite the inference model $q(s, z, c | x, y)$ using the identity $q(s, z, c | x, y) = \frac{q(s,z,c,y|x)}{q(y|x)}$, where $q(y|x)$ is the model's predictive distribution for the label $y$.

By substituting this into the ELBO expectation, we can decompose the objective into more meaningful components. Let $d... = dsdzdcdxdy$, then:

$$\mathbb{E}_{p^*(x,y)}[\mathcal{L}_{p,q}(x,y)] = \int p^*(x,y)q(c,s,z|x,y)\log\frac{p(s,z,c,x,y)}{q(s,z,c|x,y)}d...$$

$$= \int p^*(x,y)\frac{q(s,z,c,y|x)}{q(y|x)}\log\frac{p(s,z,c,x,y)q(y|x)}{q(s,z,c,y|x)}d...$$

$$= \int p^*(x,y)\frac{q(s,z,c,y|x)}{q(y|x)}\log q(y|x)d...$$

$$+ \int p^*(x,y)\frac{q(s,z,c,y|x)}{q(y|x)}\log\frac{p(s,z,c,x,y)}{q(s,z,c,y|x)}d...$$

$$= \int p^*(x)\left[p^*(y|x)\frac{\int q(z,s,c,y|x)dzdsdc}{q(y|x)}\log q(y|x)dy\right]dx+$$

$$\int p^*(x)\left[\frac{p^*(y|x)}{q(y|x)}\int q(s,z,c,y|x)\log\frac{p(s,z,c,x,y)}{q(s,z,c,y|x)}dzdsdcdy\right]dx$$

$$= \mathbb{E}_{p^*(x)}\mathbb{E}_{p^*(y|x)}[\log q(y|x)]$$

$$+ \underbrace{\mathbb{E}_{p^*(x)}\mathbb{E}_{q(s,z,c,y|x)}\left[\frac{p^*(y|x)}{q(y|x)}\log\frac{p(s,z,c,x,y)}{q(s,z,c,y|x)}\right]}_{A}$$

(9)

This decomposition yields two primary terms. The first term, $\mathbb{E}_{p^*(x,y)}[\log q(y|x)]$, is the standard cross-entropy loss for the classification task. The second term, which we denote as Term A, acts as a regularizer that aligns our inference model $q$ with the generative model $p$.

We can further simplify Term A. Based on our causal graph, we assume the factorization $q(s,z,c,y|x) = q(s,z|x)q(c|s,z)q(y|c)$ and $p(s,z,c,x,y) = p(y|c)p(c)p(s,z|c)p(x|s,z)$. This allows us to rewrite Term A as:

$$A = \mathbb{E}_{p^*(x)}\mathbb{E}_{q(s,z,c,y|x)}\left[\frac{p^*(y|x)}{q(y|x)}\log\frac{p(s,z,c,x,y)}{q(s,z,c,y|x)}\right]$$

$$= \mathbb{E}_{p^*(x)}\mathbb{E}_{q(s,z,c,y|x)}\left[\frac{p^*(y|x)}{q(y|x)}\log\frac{p(y|c)p(c)p(s,z|c)p(x|s,z)}{q(s,z|x)q(c|s,z)q(y|c)}\right]$$

$$= \mathbb{E}_{p^*(x)}\left[\int q(s,z|x)q(c|s,z)q(y|c)\frac{p^*(y|x)}{q(y|x)}\log\frac{p(s,z,c,x)}{q(s,z,c|x)}dsdzdcdy\right]$$

$$= \mathbb{E}_{p^*(x)}\left[\int\frac{p^*(y|x)}{q(y|x)}\left(\int q(s,z|x)q(c|s,z)q(y|c)\log\frac{p(s,z,c,x)}{q(s,z,c|x)}dsdzdc\right)dy\right]$$

$$= \mathbb{E}_{p^*(x,y)}\left[\frac{1}{q(y|x)}\mathbb{E}_{q(s,z,c|x)}\left[q(y|c)\log\frac{p(s,z,c,x)}{q(s,z,c|x)}\right]\right]$$

(10)

Putting it all together, and factorizing the generative model $p$ according to our causal assumptions as $p(s,z,c,x) = p(c)p(s,z|c)p(x|s,z)$, we arrive at the final learning objective:

$$\mathbb{E}_{p^*(x)}\mathbb{E}_{p^*(y|x)}[\log q(y|x)]$$

$$+ \mathbb{E}_{p^*(x,y)}\left[\frac{1}{q(y|x)}\mathbb{E}_{q(s,z,c|x)}\left[q(y|c)\log\frac{p(c)p(s,z|c)p(x|s,z)}{q(s,z|x)q(c|s,z)}\right]\right]$$

(11)

This objective function is then optimized for each client and aggregated on the server as described in the main text.

## B.1 ANALYSIS AND INTERPRETATION OF TERM A IN THE OBJECTIVE FUNCTION

The second term in our overall learning objective (equation 11), denoted as Term A, is the key term that ensures the consistency of our model's causal structure:

$$A = \mathbb{E}_{p^*(x,y)}\left[\frac{1}{q(y|x)}\mathbb{E}_{q(s,z,c|x)}\left[q(y|c)\log\frac{p(c)p(s,z|c)p(x|s,z)}{q(s,z|x)q(c|s,z)}\right]\right]$$

(12)

To reveal the theoretical properties of this term during the model optimization process, we analyze and transform it as follows:

$$
\begin{aligned}
A &= \mathbb{E}_{p^*(x,y)} \left[ \frac{1}{q(y|x)} \mathbb{E}_{q(s,z,c|x)} \left[ q(y|c) \log \frac{p(c)p(s,z|c)p(x|s,z)}{q(s,z|x)q(c|s,z)} \right] \right] \\
&= \mathbb{E}_{p^*(x,y)} \left[ \frac{1}{q(y|x)} \mathbb{E}_{q(s,z,c|x)} \left[ q(y|c) \left( \log p(x|s,z) + \log \frac{p(s,z|c)}{q(s,z|x)} + \log \frac{p(c)}{q(c|s,z)} \right) \right] \right] \\
&= \int p^*(x) \frac{p^*(y|x)}{q(y|x)} q(s,z|x)q(c|s,z)q(y|c) \\
&\quad \left( \log p(x|s,z) + \log \frac{p(s,z|c)}{q(s,z|x)} + \log \frac{p(c)}{q(c|s,z)} \right) ds dz dc dx dy \\
&\overset{a}{=} \int p^*(x) q(s,z|x)q(c|s,z)q(y|c) \log p(x|s,z) ds dz dc dx dy \\
&\quad + \int p^*(x) q(s,z|x)q(c|s,z)q(y|c) \log \frac{p(s,z|c)}{q(s,z|x)} ds dz dc dx dy \\
&\quad + \int p^*(x) q(s,z|x)q(c|s,z)q(y|c) \log \frac{p(c)}{q(c|s,z)} ds dz dc dx dy \\
&= \mathbb{E}_{p^*(x)} [\mathbb{E}_{q(s,z|x)} \log p(x|s,z)] + \mathbb{E}_{q(s,z|x)} \log \frac{p(s,z|c)}{q(s,z|x)} + \mathbb{E}_{q(c|s,z)} \log \frac{p(c)}{q(c|s,z)}
\end{aligned}
\tag{13}
$$

Step (a) above is based on Equation 11, where $q(y|x)$ is optimized to approximate the true conditional data distribution $p^*(x,y)$.

Where,

$$
\mathbb{E}_{q(s,z|x)} \log \frac{p(s,z|c)}{q(s,z|x)} = -D_{KL}(q(s,z|x)||p(s,z|c))
$$

and

$$
\mathbb{E}_{q(c|s,z)} \log \frac{p(c)}{q(c|s,z)} = -D_{KL}(q(c|s,z)||p(c))
$$

Due to the non-negativity of the $KL$ divergence ($D_{KL}(\cdot||\cdot) \geq 0$), therefore,

$$
\begin{aligned}
A &= \mathbb{E}_{p^*(x)}[\mathbb{E}_{q(s,z|x)} \log p(x|s,z)] + \mathbb{E}_{q(s,z|x)} \log \frac{p(s,z|c)}{q(s,z|x)} + \mathbb{E}_{q(c|s,z)} \log \frac{p(c)}{q(c|s,z)} \\
&= \mathbb{E}_{p^*(x)}[\mathbb{E}_{q(s,z|x)} \log p(x|s,z)] - D_{KL}(q(s,z|x)||p(s,z|c)) - D_{KL}(q(c|s,z)||p(c)) \\
&\leq \mathbb{E}_{p^*(x)}[\mathbb{E}_{q(s,z|x)} \log p(x|s,z)] \\
&\leq \mathbb{E}_{p^*(x)}[\log \mathbb{E}_{q(s,z|x)} p(x|s,z)] \leq \mathbb{E}_{p^*(x)}[\log p(x)]
\end{aligned}
\tag{14}
$$

This derivation explicitly shows that the objective of maximizing Term A is theoretically aligned with the objective of maximizing the marginal log-likelihood of the data, $\log p(x)$, thus proving the effectiveness of this regularization term.

# C  DATASET DETAILS

## C.1  DATASETS

To comprehensively evaluate the performance of our model, we designed two core categories of experiments: generalization to unseen domains and robustness to data shifts. To this end, we selected a series of benchmark datasets covering both natural images and specialized medical images.

### C.1.1  EVALUATION OF GENERALIZATION TO UNSEEN DOMAINS.

In this section, we aim to test the model's performance when encountering data from entirely new distributions (i.e., "unseen domains" or "unseen clients"). We selected the following four datasets, which are widely used in domain generalization research.

- **PACS:** This is a classic benchmark dataset for domain generalization in image classification. It consists of four domains with distinct styles: Photo (P), Art Painting (A), Cartoon (C), and Sketch (S). These four domains share the same 7 object categories. Due to the significant visual style differences between the domains (i.e., a large domain gap), PACS serves as an ideal choice for evaluating whether the knowledge learned from known domains can be generalized to unknown styles.

- **Office-Home:** This is a larger-scale image classification benchmark designed for domain adaptation and generalization tasks. It contains approximately 15,500 images covering 65 categories of everyday objects, distributed across four visually distinct domains: Artistic, Clipart, Product, and Real-World. For each domain within this dataset, we strictly partition the data into a 60% training set, a 20% validation set, and a 20% test set to ensure fairness and consistency in our evaluation.

- **Brain Tumor (BT):** To extend our evaluation to the critical domain of medical imaging, we adopted this public Kaggle dataset. The dataset consists of images based on brain magnetic resonance imaging scans, classified into four diagnostic categories: glioma tumor, meningioma tumor, pituitary Tumor, and no Tumor. In this context, different hospitals, scanning devices, or patient demographics can be considered as distinct "domains," making it an excellent use case for testing the model's generalization capabilities in realistic medical environments.

- **Modern-Office31:** This is a refined variant of the classic Office-31 dataset, which increases the complexity of the challenge by introducing synthetic data. This dataset focuses on 31 categories related to office supplies and includes four domains: Amazon images collected from e-commerce websites, images captured by a low-resolution Webcam, images taken with a high-resolution DSLR camera, and an algorithmically generated Synthetic image domain. For data within each domain, we follow an 80% for training and 20% for testing split.

## C.2 Simulating Real-World Non-IID Client Data

To simulate the non-Independent and Identically Distributed (Non-IID) scenarios of real-world FL, we partition the entire original training and testing datasets among all clients. The core of this data partitioning strategy is the use of a Dirichlet distribution to control the data composition for each client (Hsu et al., 2019).

Specifically, for each client $k$, we sample the proportion of samples for each class $j$, denoted as $p_{j,k}$, from a Dirichlet distribution: $p_{j,k} \sim \text{Dir}(\alpha)$. In this setup, the hyperparameter $\alpha$ precisely controls the degree of data heterogeneity among clients. A smaller value of $\alpha$ results in stronger data heterogeneity, meaning greater discrepancies in data distribution across clients. Conversely, a larger $\alpha$ value leads to a more uniform data distribution.

In our specific experimental settings, we adopted the following strategies for different datasets:

- For the PACS dataset: This dataset comprises multiple domains, such as Art Painting, Cartoon, Photo, and Sketch. We evenly distribute the data from each domain among 5 clients to simulate clients from distinct data sources. The remaining unseen domains are held out to simulate "unseen clients" not encountered during the training process.

- For the CIFAR-10/100 datasets: We construct varying degrees of data heterogeneity by setting different values for $\alpha$. This allows for a comprehensive evaluation of the model's performance under different levels of Non-IID conditions.

## D  Experimental Implementation Details

### D.1  Implementation Details

All our experiments were conducted under a unified federated learning framework. The global training process consists of 50 communication rounds. In each round, clients perform local updates, after which their model parameters are sent back to the server for aggregation. All experiments

were implemented on an *NVIDIA GeForce RTX 3090 GPU*, with a software environment based on *PyTorch 2.1.2* and *Python 3.10*.

**Configuration for Cross-Domain Generalization Experiments:** When evaluating the model's performance on unseen domain clients (using datasets such as PACS and Office-Home), we employed a pre-trained *ViT-B/32* model as the backbone for the image encoder. Throughout the training process, we froze the parameters of the CLIP encoder to fully leverage its powerful pre-trained features. For the local training on clients, we used the *Adam* optimizer with hyperparameters set to $\beta_1 = 0.9$ and $\beta_2 = 0.98$, and a weight decay of $0.02$. The local learning rate was fixed at $5e - 5$, and each client performed 5 local epochs of training with a batch size of 32 in each communication round.

**Configuration for Covariate Shift Robustness Experiments:** To assess the model's capability against covariate shift (i.e., on the CIFAR-10-C/100-C datasets), we selected a WideResNet **?** as the feature extraction model. During the local training phase on clients, we also set the number of local training epochs to 5. Model optimization was carried out using the SGD optimizer. The learning rate for both the feature encoding model and our proposed causal model was uniformly set to 0.001.

Table 5: Leave-Two-Domain-Out cross-validation accuracy (%) on the PACS based on a dirichlet distribution. $C_1 - C_5$ and $C_6 - C_{10}$ are groups of clients from two different source domains.

| Method | Source | | | | | | | | | | Target | | Avg |
|---|---|---|---|---|---|---|---|---|---|---|---|---|---|
| | S | | | | | A | | | | | C | P | |
| | $C_1$ | $C_2$ | $C_3$ | $C_4$ | $C_5$ | $C_6$ | $C_7$ | $C_8$ | $C_9$ | $C_{10}$ | | | |
| FedCLIP | $85.24_{\pm2.0}$ | $88.89_{\pm3.9}$ | $86.98_{\pm0.7}$ | $82.70_{\pm1.8}$ | $76.28_{\pm0.9}$ | $90.00_{\pm0.4}$ | $79.49_{\pm3.6}$ | $83.33_{\pm2.1}$ | $97.54_{\pm0.6}$ | $100.0_{\pm0.1}$ | $94.99_{\pm0.5}$ | $97.46_{\pm0.7}$ | 88.57 |
| FedProx | $92.46_{\pm0.7}$ | $86.67_{\pm0.0}$ | $90.31_{\pm2.2}$ | $87.62_{\pm1.4}$ | $91.79_{\pm0.9}$ | $89.46_{\pm0.0}$ | $83.67_{\pm3.6}$ | $87.31_{\pm3.6}$ | $84.62_{\pm2.1}$ | $88.79_{\pm5.9}$ | $91.46_{\pm1.2}$ | $92.19_{\pm1.4}$ | 88.86 |
| FedAvg | $90.93_{\pm2.9}$ | $89.45_{\pm3.9}$ | $92.92_{\pm1.5}$ | $87.36_{\pm1.1}$ | $93.08_{\pm0.6}$ | $87.93_{\pm1.0}$ | $86.45_{\pm3.6}$ | $89.92_{\pm2.1}$ | $84.36_{\pm0.3}$ | $90.08_{\pm1.2}$ | $91.56_{\pm1.0}$ | $92.96_{\pm1.2}$ | 89.75 |
| MOON | $91.69_{\pm2.2}$ | $83.89_{\pm3.9}$ | $93.44_{\pm1.3}$ | $88.39_{\pm1.8}$ | $92.44_{\pm0.9}$ | $88.69_{\pm0.0}$ | $80.89_{\pm0.0}$ | $90.44_{\pm0.0}$ | $85.39_{\pm0.5}$ | $89.44_{\pm0.0}$ | $91.20_{\pm1.6}$ | $92.46_{\pm0.1}$ | 89.03 |
| FAACLIP | $92.89_{\pm1.2}$ | $88.89_{\pm3.9}$ | $89.06_{\pm2.2}$ | $84.73_{\pm2.9}$ | $88.55_{\pm0.0}$ | $90.0_{\pm0.0}$ | $76.92_{\pm0.0}$ | $90.36_{\pm3.2}$ | $97.32_{\pm0.6}$ | $95.83_{\pm2.9}$ | $96.33_{\pm0.3}$ | $98.76_{\pm0.3}$ | 90.82 |
| Ours | $93.13_{\pm1.2}$ | $88.89_{\pm3.9}$ | $94.79_{\pm1.5}$ | $87.28_{\pm2.5}$ | $86.54_{\pm1.6}$ | $93.33_{\pm4.7}$ | $94.87_{\pm3.6}$ | $90.91_{\pm2.1}$ | $97.32_{\pm0.0}$ | $97.74_{\pm0.0}$ | $97.63_{\pm0.3}$ | $99.70_{\pm0.0}$ | 93.51 |
| | P | | | | | S | | | | | C | A | |
| | $C_1$ | $C_2$ | $C_3$ | $C_4$ | $C_5$ | $C_6$ | $C_7$ | $C_8$ | $C_9$ | $C_{10}$ | | | |
| FedCLIP | $100.0_{\pm0.0}$ | $96.97_{\pm0.5}$ | $97.96_{\pm1.0}$ | $97.06_{\pm1.4}$ | $100.0_{\pm0.0}$ | $83.46_{\pm1.6}$ | $86.11_{\pm3.9}$ | $85.94_{\pm5.3}$ | $80.91_{\pm1.1}$ | $75.64_{\pm0.9}$ | $94.09_{\pm0.4}$ | $95.9_{\pm0.3}$ | 91.17 |
| FedProx | $98.68_{\pm2.1}$ | $96.88_{\pm1.4}$ | $93.64_{\pm1.0}$ | $93.04_{\pm1.4}$ | $97.91_{\pm0.0}$ | $85.55_{\pm0.9}$ | $87.06_{\pm0.7}$ | $85.64_{\pm3.6}$ | $85.04_{\pm0.6}$ | $86.93_{\pm0.9}$ | $83.28_{\pm0.8}$ | $85.50_{\pm0.6}$ | 89.93 |
| FedAvg | $98.68_{\pm2.1}$ | $96.88_{\pm0.0}$ | $93.64_{\pm1.0}$ | $92.06_{\pm1.4}$ | $95.00_{\pm0.6}$ | $84.10_{\pm0.4}$ | $86.93_{\pm0.9}$ | $84.96_{\pm1.0}$ | $84.06_{\pm0.7}$ | $87.03_{\pm0.9}$ | $80.14_{\pm1.0}$ | $82.76_{\pm0.3}$ | 88.86 |
| MOON | $100.0_{\pm0.0}$ | $96.88_{\pm1.2}$ | $93.64_{\pm1.0}$ | $93.04_{\pm1.4}$ | $95.00_{\pm0.5}$ | $87.00_{\pm3.2}$ | $86.33_{\pm1.3}$ | $85.64_{\pm0.0}$ | $85.04_{\pm1.0}$ | $87.00_{\pm0.9}$ | $82.21_{\pm0.2}$ | $85.01_{\pm0.4}$ | 89.73 |
| FAACLIP | $100.0_{\pm0.0}$ | $96.97_{\pm0.0}$ | $96.32_{\pm1.0}$ | $98.02_{\pm1.4}$ | $100.0_{\pm0.0}$ | $82.44_{\pm4.7}$ | $88.89_{\pm3.9}$ | $85.94_{\pm2.9}$ | $84.48_{\pm2.9}$ | $79.49_{\pm3.6}$ | $96.47_{\pm0.2}$ | $94.86_{\pm0.1}$ | 91.99 |
| Ours | $100.0_{\pm0.0}$ | $100.0_{\pm0.0}$ | $97.96_{\pm0.0}$ | $97.06_{\pm0.9}$ | $100.0_{\pm0.0}$ | $93.38_{\pm2.9}$ | $86.11_{\pm3.9}$ | $94.27_{\pm1.5}$ | $86.26_{\pm1.6}$ | $83.33_{\pm2.4}$ | $97.65_{\pm0.3}$ | $96.37_{\pm0.5}$ | 94.37 |

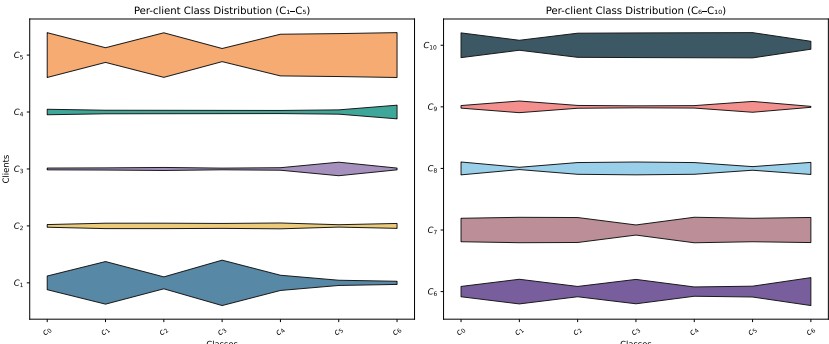

Figure 8: An illustrative example of the simulated data distribution for the "kite" class from the PACS dataset across different clients. In the figure, $C_1$ to $C_5$ represent five clients, and $c_1$ to $c_7$ represent seven subclasses.

## D.2 GENERALIZATION TO MULTIPLE UNSEEN DOMAINS (PACS)

**Experimental Design:** The experiment employs a rigorous Leave-Two-Domain-Out Cross-validation methodology on the classic domain generalization benchmark, the PACS dataset. It simulates a non-IID federated learning environment where ten groups of clients ($C_1$ to $C_{10}$) source their data from only two of the four PACS domains (typically Photo (P), Art Painting (A), Cartoon (C), and Sketch (S)), designated as "Source Domains." The data allocation follows a Dirichlet distribution to model the stylistic and content-based heterogeneity among clients. After training, the

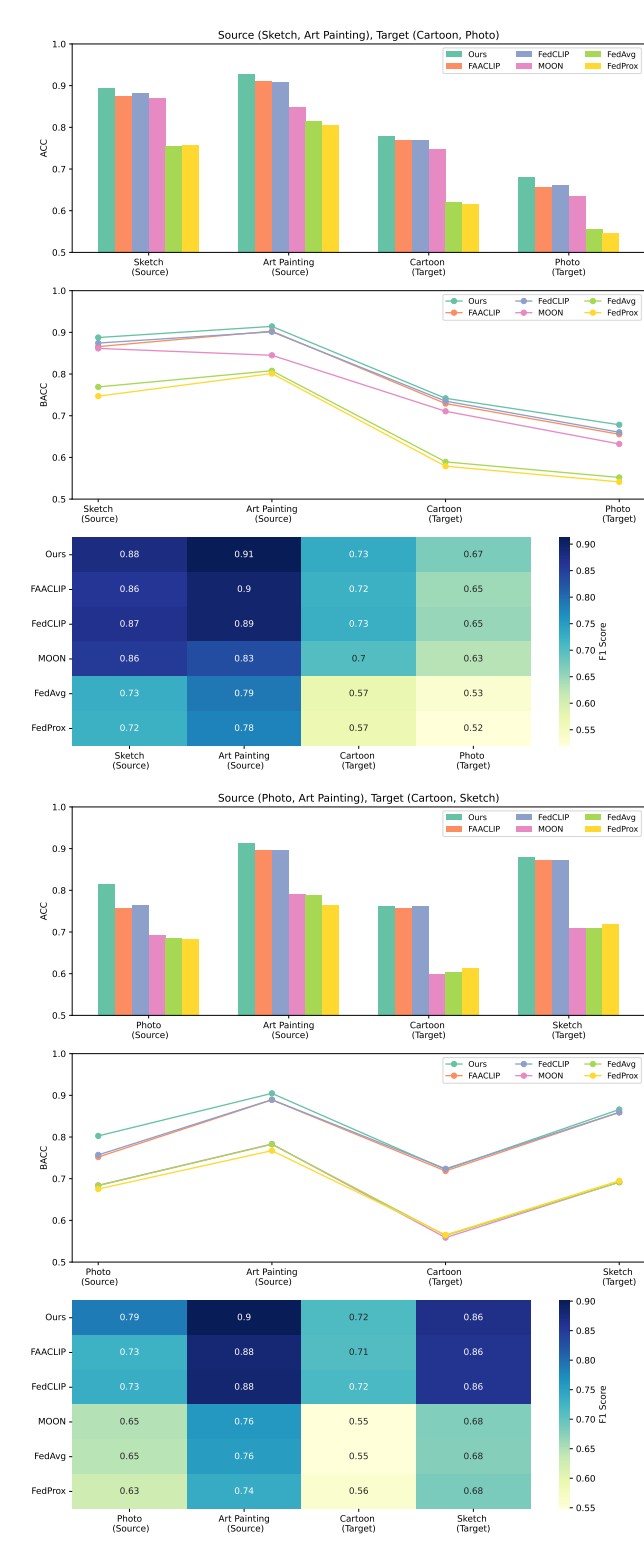

Figure 9: Performance Comparison of Federated Models Across Classes on ModernOffice31. Left: Sketch and Art Painting as source domains, Cartoon and Photo as target. Right: Photo and Art Painting as source domains, Cartoon and Photo as target.

model's performance is evaluated not only on the source domains but, more critically, on the two entirely unseen "Target Domains," which directly measures its generalization capability.

**Results Analysis:** The table 5 is divided into two sections, each presenting the test results from a different combination of source and target domains.

- **Overall Performance:** Looking at the final average accuracy (Avg) column, our model (Ours) achieves the leading overall performance in both experimental setups. In the top half of the table, our model reaches a mean accuracy of 93.51%, significantly outperforming all baseline methods, such as the runner-up FAACLIP (90.82%) and the conventional FedAvg (89.75%). Similarly, in the bottom half, our model once again takes the lead with an accuracy of 94.37%, demonstrating its consistent superiority.

- **Generalization Ability:** The most critical metric in this experiment is the performance on the Target domains. In this aspect, our model performs exceptionally well. For instance, in the first experimental setup (top half), our model achieves impressive accuracies of 97.63% and 99.70% on the two unseen domains, C and P, respectively. In the second setup (bottom half), it also obtains high scores of 97.65% and 96.37% on the unseen C and A domains. This provides strong evidence that our model can effectively learn generalizable features from the source domains, rather than merely overfitting to their superficial statistical patterns.

In summary, this experiment, conducted under the challenging "leave-two-domain-out" setting, systematically demonstrates the powerful generalization capability of our model. Compared to other algorithms, our model is not only robust on the known source domains but, more importantly, can more successfully transfer its knowledge to entirely new, unseen data distributions, showcasing robustness and high accuracy that far exceed existing methods.

Table 6: Leave-Two-Domain-Out generalization accuracy (%) on PACS.

| Method | Source | | | | | | | | | | Target | | Avg |
|---|---|---|---|---|---|---|---|---|---|---|---|---|---|
| | A | | | | | C | | | | | | | |
| | $C_1$ | $C_2$ | $C_3$ | $C_4$ | $C_5$ | $C_6$ | $C_7$ | $C_8$ | $C_9$ | $C_{10}$ | P | S | |
| FedCLIP | 98.66 | 92.31 | 100.0 | 100.0 | 97.87 | 100.0 | 84.62 | 90.91 | 96.64 | 87.5 | 98.34 | 84.19 | 94.28 |
| FedAVG | 97.32 | 92.31 | 100.0 | 100.0 | 95.74 | 90.0 | 84.62 | 81.82 | 95.97 | 100.0 | 97.66 | 82.61 | 93.17 |
| FAACLIP | 97.32 | 92.31 | 100.0 | 100.0 | 94.01 | 90.0 | 84.62 | 90.91 | 96.64 | 100.0 | 98.70 | 85.31 | 94.15 |
| Ours | 97.32 | 92.31 | 100.0 | 100.0 | 97.87 | 90.0 | 84.62 | 95.45 | 97.32 | 100.0 | **99.64** | **86.43** | **95.08** |
| | C | | | | | P | | | | | | | |
| | $C_1$ | $C_2$ | $C_3$ | $C_4$ | $C_5$ | $C_6$ | $C_7$ | $C_8$ | $C_9$ | $C_{10}$ | S | A | |
| FedCLIP | 97.99 | 92.31 | 100.0 | 100.0 | 95.74 | 95.65 | 100.0 | 97.96 | 97.06 | 100.0 | 83.79 | 96.19 | 96.39 |
| FedAVG | 98.66 | 92.31 | 100.0 | 100.0 | 96.14 | 91.65 | 83.34 | 97.96 | 97.06 | 100.0 | 81.64 | 92.99 | 94.31 |
| FAACLIP | 96.64 | 92.31 | 100.0 | 100.0 | 94.01 | 95.84 | 100.0 | 97.96 | 97.06 | 100.0 | 83.45 | 94.75 | 96.00 |
| Ours | 96.64 | 92.31 | 100.0 | 100.0 | 97.87 | 95.65 | 100.0 | 97.96 | 97.06 | 100.0 | **85.53** | **96.90** | **96.66** |
| | C | | | | | S | | | | | | | |
| | $C_1$ | $C_2$ | $C_3$ | $C_4$ | $C_5$ | $C_6$ | $C_7$ | $C_8$ | $C_9$ | $C_{10}$ | P | A | |
| FedCLIP | 97.32 | 92.31 | 100.0 | 100.0 | 96.14 | 92.95 | 83.33 | 95.31 | 92.37 | 86.54 | 98.40 | 95.14 | 94.15 |
| FedAVG | 97.32 | 92.31 | 100.0 | 100.0 | 95.74 | 80.92 | 83.33 | 85.94 | 79.39 | 76.92 | 97.66 | 94.91 | 90.36 |
| FAACLIP | 96.64 | 92.31 | 100.0 | 100.0 | 96.14 | 92.95 | 83.33 | 96.88 | 93.13 | 86.54 | 98.76 | 95.34 | **94.33** |
| Ours | 97.99 | 92.31 | 100.0 | 100.0 | 97.87 | 93.13 | 83.33 | 95.31 | 86.26 | 82.69 | **99.70** | **96.48** | 93.76 |

The experiment in Table 6 involves three cross-domain scenarios: using art (A) and cartoon (C) as source domains with photo (P) and sketch (S) as target domains; using C and P as source domains with S and A as target domains; and using C and S as source domains with P and A as target domains. Each method was trained on 10 clients ($C_1 - C_{10}$) and evaluated separately on the target domains.

The results demonstrate that the proposed method (Ours) achieves the best performance on the target domains: in the first scenario, it reaches 99.64% on domain P and 86.43% on domain S; in the second scenario, it achieves 85.53% on domain S and 96.90% on domain A; and in the third scenario, it attains 99.70% on domain P and 96.48% on domain A. These values are the highest among all compared methods, highlighting its exceptional cross-domain generalization capability.

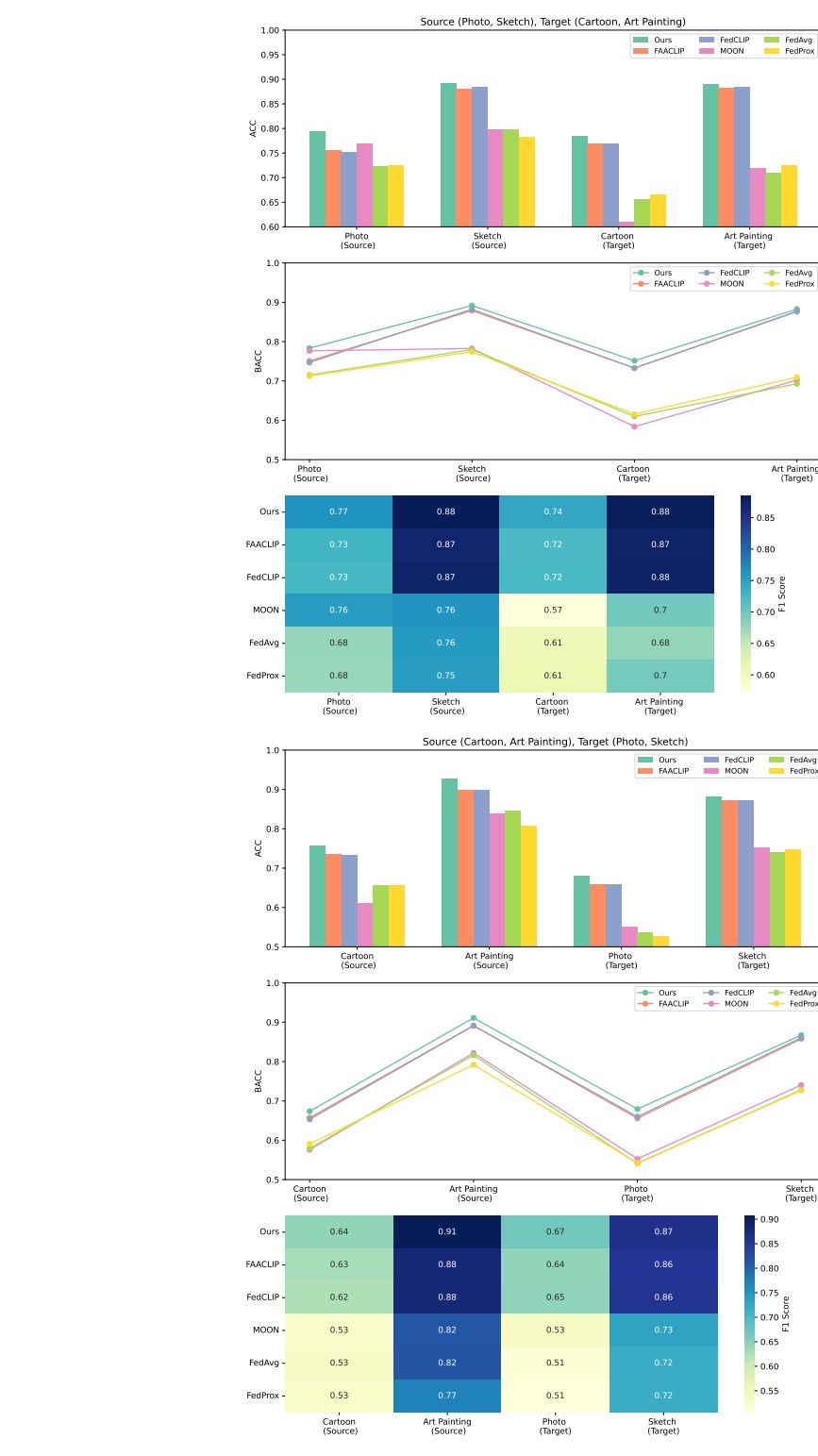

Figure 10: Performance Comparison of Federated Models Across Classes on ModernOffice31. Left: Photo and Sketch as source domains, Cartoon and Art Painting as target. Right: Cartoon and Art Painting as source domains, Photo and Sketch as target.

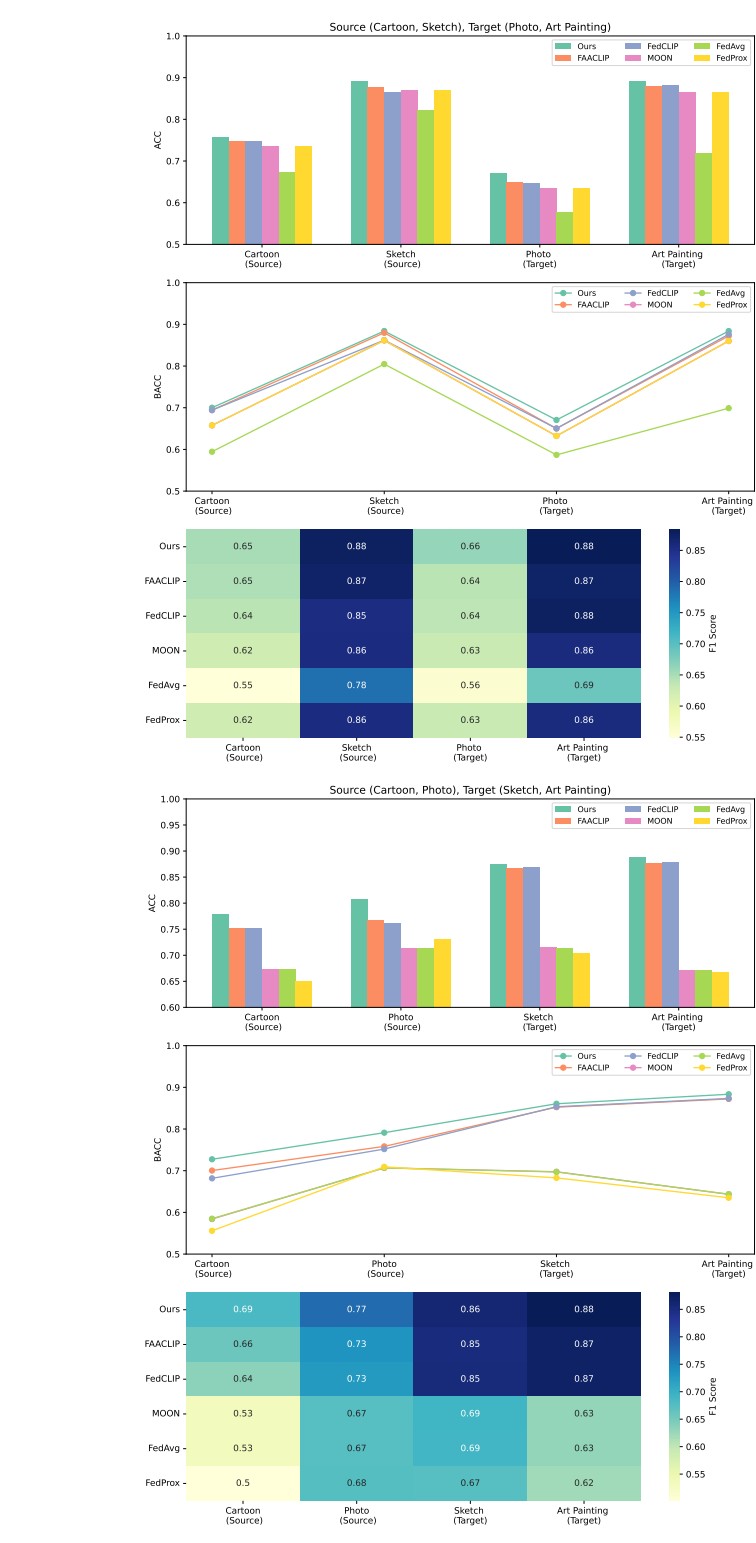

Figure 11: Performance Comparison of Federated Models Across Classes on ModernOffice31. Left: Cartoon and Sketch as source domains, Photo and Art Painting as target. Right: Cartoon and Photo as source domains, Sketch and Art Painting as target.

### D.3 GENERALIZATION TO MULTIPLE UNSEEN DOMAINS (MODERNOFFICE31)

This set of experiments (as shown in Figures 9, 10, and 11) evaluates the domain generalization capability of our model using a Leave-Two-Domain-Out cross-validation protocol. The experiments were conducted on the ModernOffice31 dataset, which contains four domains (Sketch, Art Painting, Cartoon, and Photo). In each round of testing, the model is trained using data from only two domains as the Source, and then its performance is evaluated on the other two completely unseen domains as the Target. By rotating through different combinations of source and target domains, this experimental setup comprehensively examines the ability of each algorithm to learn generalizable knowledge from limited and biased data.

Analysis of Results on Unseen Domains: When focusing on the model's performance on the Target Domains, a clear and significant advantage of our model can be observed across three key metrics: Accuracy (ACC), Balanced Accuracy (BACC), and F1 Score.

1. **Accuracy Comparison (Bar Charts):** In the bar chart section of each figure, the bars corresponding to the "Target" domains clearly display how each model performs after being introduced to a new environment. In almost all test configurations (e.g., Cartoon and Photo as target domains in the left panel of Figure 9, and Sketch and Art Painting as targets in the right panel of Figure 11), the dark green bar representing our model is consistently the highest among the target domains. This visually indicates that our model can maintain the highest classification accuracy when faced with unseen data distributions.

2. **Cross-Domain Performance Trend (BACC Line Charts):** The line charts in the middle vividly reveal the performance degradation of each model when transitioning from source to target domains, measured by BACC. BACC effectively evaluates model performance, especially under class imbalance. Most baseline methods, particularly FedAvg and FedProx, exhibit a sharp drop in their BACC curves upon entering the target domains, showing that their generalization ability is fragile against domain shifts and potential class imbalances. In contrast, while our model's curve also shows a slight decline, it consistently remains above all other curves, and its rate of decline is comparatively gentler. This demonstrates that our model possesses stronger robustness against domain shifts, with minimal performance loss.

3. **F1 Score Comparison (Heatmaps):** The heatmaps at the bottom provide the most granular performance measurement. Observing the columns corresponding to the target domains, the cells in the row for our model are consistently the darkest in color, representing the highest values. For example, in the right panel of Figure 10, when Photo and Sketch are the target domains, our model achieves F1 scores of 0.67 and 0.87, respectively, significantly outperforming all other methods. This indicates that our model is more reliable.

In summary, this comprehensive experiment, conducted under the highly challenging "leave-two-domain-out" setting, provides powerful evidence of our model's superior domain generalization capability. Compared to other algorithms, when transitioning from two source domains to two entirely new, unseen target domains, our model demonstrates a consistent and significant lead across all key performance metrics. This proves its ability to more effectively learn core features that are generalizable and independent of any specific domain.

### D.4 GENERALIZATION TO A SINGLE UNSEEN DOMAIN

Table 7 presents a comparative experiment conducted on the domain generalization dataset, ModernOffice31. This experiment follows the Leave-One-Domain-Out cross-validation protocol. The ModernOffice31 dataset comprises four distinct domains. In each round of the experiment, three of these domains are selected as "Source Domains for training, while the remaining one is used as a completely unseen "Target Domain for testing. This process is repeated four times, ensuring that each domain serves as the target domain once, thereby comprehensively evaluating the ability of each model (including our model "Ours" and six other baseline methods like FedCLIP and FedProx) to learn from multi-source data and generalize to unknown environments.

The four sections of the table correspond to the results of the four cross-validation rounds, with the data clearly demonstrating the superiority of our model.

| Method | Source | | | Target | Avg |
|---|---|---|---|---|---|
| | A | D | S | W | |
| FedCLIP | 91.82 | 94.94 | 65.00 | 88.42 | 85.04 |
| FedProx | 82.41 | 81.82 | 74.03 | 79.49 | 79.43 |
| FedAVG | 59.07 | 69.38 | 59.67 | 64.53 | 63.16 |
| MOON | 84.01 | 81.82 | 74.68 | 83.14 | 80.91 |
| PromptFL | 90.05 | 78.79 | 79.52 | 68.23 | 79.15 |
| FAA-CLIP | 95.38 | **98.99** | 82.74 | **91.95** | 92.26 |
| Ours | **95.55** | 95.92 | **85.42** | 91.06 | **91.99** |
| | D | S | W | A | |
| FedCLIP | 95.95 | 65.32 | 88.05 | 90.06 | 84.84 |
| FedProx | 89.90 | 59.52 | 90.57 | 66.88 | 76.71 |
| FedAVG | 88.89 | 49.19 | 91.19 | 42.81 | 68.02 |
| MOON | 95.96 | 65.81 | 91.19 | 66.92 | 79.97 |
| FAA-CLIP | 100.0 | 80.32 | 94.34 | 92.19 | 91.71 |
| Ours | 97.96 | 78.71 | **94.54** | **92.51** | 90.93 |
| | S | W | A | D | |
| FedCLIP | 64.67 | 86.16 | 91.82 | 89.15 | 82.95 |
| FedProx | 73.55 | 90.57 | 81.53 | 89.36 | 83.75 |
| FedAVG | 76.94 | 93.08 | 82.59 | 92.17 | 86.19 |
| MOON | 75.00 | 92.45 | 87.39 | 85.74 | 85.14 |
| PromptFL | 79.68 | 77.99 | 90.41 | 67.71 | 78.95 |
| FAA-CLIP | 81.93 | 93.71 | 95.38 | 95.98 | 91.75 |
| Ours | **84.84** | 92.28 | **95.73** | **96.58** | **92.36** |
| | W | A | D | S | |
| FedCLIP | 87.42 | 91.47 | 95.95 | 54.77 | 82.40 |
| FedProx | 91.82 | 81.17 | 89.90 | 46.45 | 77.33 |
| FedAVG | 95.60 | 80.46 | 97.98 | 45.87 | 79.97 |
| MOON | 94.34 | 82.06 | 89.90 | 54.77 | 80.27 |
| PromptFL | 81.76 | 88.45 | 85.86 | 32.49 | 72.14 |
| FAA-CLIP | 96.23 | 94.85 | 98.99 | 57.10 | 86.79 |
| Ours | 96.21 | 94.65 | 95.92 | 56.82 | 85.90 |

Table 7: Accuracy(%) in the ModernOffice31 dataset. Bold means the best.

- Overall Performance: Based on the final average accuracy (Avg), our model performs best in the vast majority of cases. Across the four experimental configurations, our model achieved the highest overall average score in three of them, with figures of 91.99%, 90.93%, and 92.36%, respectively. This indicates that our model can maintain top-tier and robust overall performance regardless of the source and target domain combination.

- Generalization to Target Domain: The most critical metric for measuring domain generalization capability is the accuracy on the unseen "Target Domain". Our model also shows outstanding performance in this regard. For instance, in the third section of the experiment, when the unseen domain "D" was the target, our model achieved the highest accuracy of 96.58%. In the first section, although FAA-CLIP held a slight advantage on the target domain W, our model surpassed it in overall average score, proving its stronger holistic robustness.

This experiment systematically demonstrates the powerful domain generalization capabilities of our model. Compared to other algorithms, our model not only learns effectively from multiple known source domains but, more importantly, can efficiently transfer this learned knowledge and apply it to entirely new, unseen data distributions, consistently maintaining top-level prediction accuracy and stability in varying test environments.

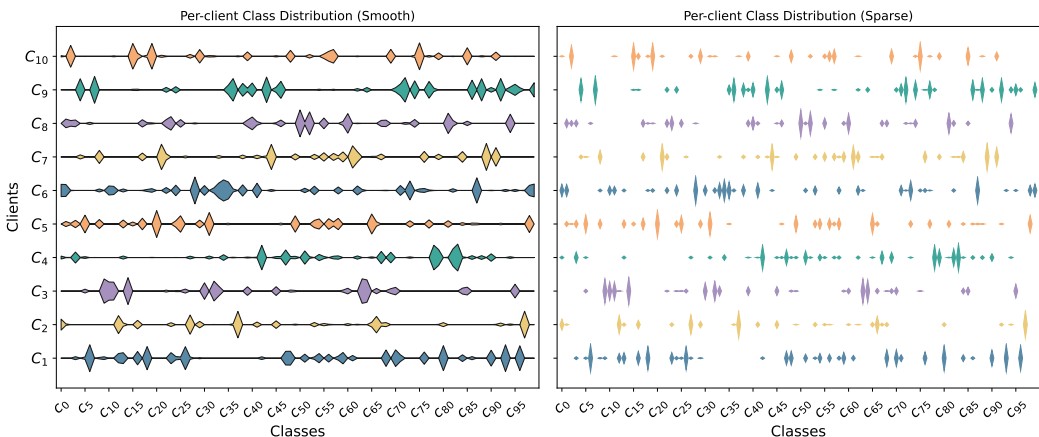

Figure 12: An illustrative example of the simulated data distribution for the "kite" class from the CIFAR100 dataset across different clients. In the figure, $C_1$ to $C_{10}$ represent ten clients, and $c_0$ to $c_{100}$ represent classes.

| Corruption Type | FedAvg | FedLN | FedIIR | FedCLIP | FOOGD | FAA-CLIP | Ours |
|---|---|---|---|---|---|---|---|
| None | 68.03 | 75.24 | 68.26 | 72.92 | 75.09 | 81.26 | **86.83** |
| Brightness | 65.73 | 71.77 | 66.12 | 69.04 | 73.71 | 80.39 | **85.00** |
| Fog | 53.89 | 60.82 | 54.85 | 62.68 | 60.96 | 69.70 | **73.79** |
| Frosted glass blur | 43.13 | 42.33 | 44.53 | 68.86 | 45.80 | 68.77 | **77.19** |
| Motion blur | 41.30 | 52.65 | 44.23 | 68.96 | 51.05 | 69.36 | **77.69** |
| Snow | 54.80 | 60.55 | 55.52 | 67.97 | 61.90 | 71.60 | **80.19** |
| Contrast | 41.25 | 45.02 | 41.35 | 56.31 | 49.14 | 58.51 | **68.41** |
| Frost | 56.21 | 58.22 | 55.91 | 64.63 | 63.84 | 69.24 | **78.00** |
| Impulse noise | 49.32 | 50.52 | 48.45 | 67.47 | 52.33 | 67.82 | **74.51** |
| Pixelate | 56.88 | 62.00 | 59.10 | 70.70 | 64.37 | 79.21 | **80.05** |
| Defocus blur | 52.37 | 61.08 | 52.72 | 69.93 | 58.66 | 74.40 | **80.47** |
| Jpeg compression | 61.56 | 68.61 | 60.46 | 70.78 | 66.55 | 78.46 | **84.36** |
| Elastic transform | 52.12 | 61.29 | 53.21 | 68.99 | 59.18 | 73.87 | **81.97** |
| Gaussian Noise | 48.66 | 50.25 | 49.15 | 69.20 | 53.92 | 69.77 | **71.03** |
| Shot noise | 52.73 | 54.55 | 53.09 | 69.59 | 58.31 | 70.53 | **78.37** |
| Zoom blur | 45.15 | 54.88 | 46.57 | 68.29 | 52.97 | 68.71 | **78.35** |
| Spatter | 62.18 | 67.33 | 60.97 | 69.47 | 65.31 | 78.73 | **83.22** |
| Gaussian blur | 46.86 | 55.64 | 47.51 | 68.99 | 53.26 | 70.38 | **71.03** |
| Saturate | 63.62 | 71.76 | 63.32 | 68.33 | 71.98 | 80.36 | **84.73** |
| Speckle noise | 52.25 | 54.30 | 53.20 | 69.81 | 57.72 | 70.35 | **77.92** |
| **Avg.** | 53.40 | 58.94 | 53.93 | 68.15 | 59.8 | 72.57 | **78.66** |

Table 8: Generalization Performance of Federated Learning Methods on CIFAR-10-C (Trained on CIFAR-10 with $\alpha = 0.1$)– Vertical Comparison

## D.5 PERFORMANCE EVALUATION UNDER COVARIATE SHIFT

The experimental results in Table 8 clearly demonstrate the exceptional robustness of the "Ours" method within a federated learning environment. The data shows that for all 19 corruption types listed, as well as for the original, uncorrupted test set, the classification accuracy of the "Ours" method is significantly superior to the other six comparative methods, with a particularly pronounced performance advantage on corruption types such as "JPEG Compression," "Saturate," and "Spatter." This is further substantiated by the final average performance (Avg.). The "Ours" method achieved a high average accuracy of 78.66%, far surpassing the runner-up, FAA-CLIP (72.57%), and marking a massive improvement of over 25 percentage points compared to the classic baseline algorithm,

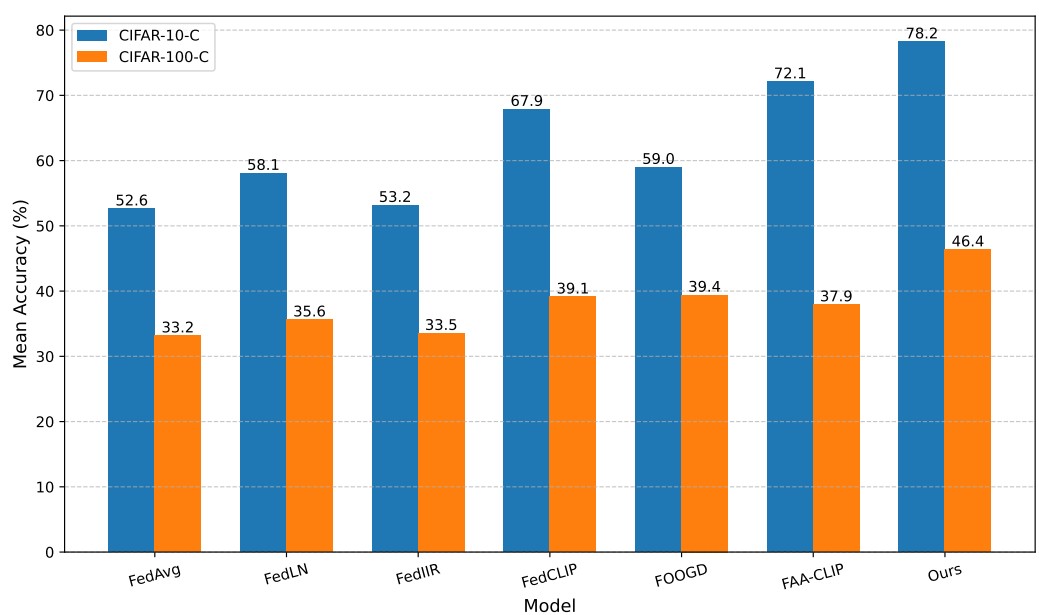

Figure 13: Model Performance on CIFAR-10-C and CIFAR-100-C.

FedAvg (53.4%). These figures provide compelling evidence that the proposed method not only excels under ideal conditions but also displays powerful generalization capabilities and stability when confronted with various simulated real-world data corruptions, highlighting its significant potential for practical applications.

Figure 13 presents a detailed performance comparison of our model (Ours) against six baseline algorithms on the CIFAR-10-C and CIFAR-100-C datasets. The reported Mean Accuracy is the average performance across various types of data corruptions (e.g., blur, noise, weather effects), which provides a comprehensive evaluation of each model's robustness. The results clearly indicate that our model achieves the best performance on both datasets. Specifically, on CIFAR-10-C, our model reached a mean accuracy of 78.2%, significantly outperforming the next-best model, FAA-CLIP (72.1%). On the more challenging CIFAR-100-C dataset, which features more classes, our model also ranked first with an accuracy of 46.4%, further widening the performance gap with the second-place FAA-CLIP (37.9%). This data provides strong evidence that, compared to existing methods, our model possesses superior generalization ability and robustness when faced with unknown distribution shifts and data corruptions.

Figure 12 shows the per-client class distribution on the CIFAR-100 dataset with a Dirichlet distribution parameter of $\alpha = 0.1$. The left subfigure illustrates a smooth distribution, while the right subfigure presents a sparse one, revealing significant differences in class distributions across clients under the Non-IID setting.

# E ABLATION STUDIES

To further investigate the performance and stability of our proposed model under different client scales, we conducted an ablation study where the total number of clients, $K$, was set to 6, 10, and 20, respectively. We compared our method with a series of baseline models: FAACLIP, FedAvg, FedCLIP, FedProx, and MOON. The experimental results are shown in Figure 14, where the left plot displays the mean accuracy (ACC, Mean ± Std) on the target domain, and the right plot shows the mean AUC values.

As can be clearly seen from both plots, our method (red line/bar) consistently and significantly outperforms all baseline models across all settings of client numbers. Our model achieves the highest performance in terms of both ACC and AUC metrics. More importantly, as shown in the left plot,

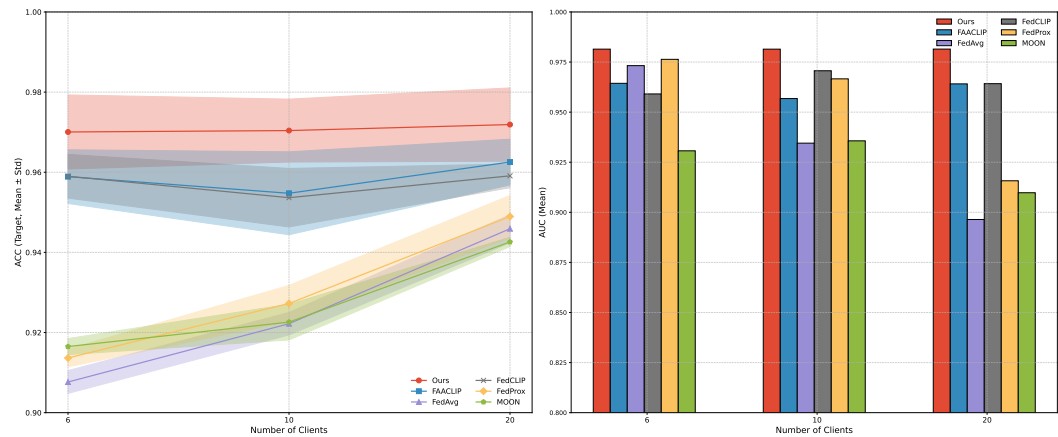

Figure 14: Impact of the Number of Clients ($K$) on Model Performance.

the performance curve of our model is very flat, and the standard deviation range (red shaded area) is extremely narrow, indicating that its performance is minimally affected by the variation in the number of clients, demonstrating high stability.

FedCLIP and FAACLIP are the next-best performing methods, yet their accuracy and AUC values still show a clear gap compared to our model. Methods such as FedAvg, FedProx, and MOON exhibit relatively poor performance. This ablation study strongly demonstrates that our proposed method surpasses existing mainstream approaches in terms of performance.

## F  RELATED WORK

### F.1  FEDERATED LEARNING

FL aims to solve data silo and privacy issues through distributed collaborative training (McMahan et al., 2017), but its core challenge lies in client drift caused by data heterogeneity, which severely impairs the performance of the global model (Xiao et al., 2024). To tackle this, existing strategies include: introducing a proximal term to constrain local updates (Li et al., 2020); retaining private batch normalization layers to adapt to local features (Li et al., 2021b); utilizing contrastive learning to align model representations (Li et al., 2021a); and building domain-invariant knowledge through a federated graph learning (Xiao et al., 2024). However, while effective against heterogeneity, these methods often fail under complex domain or covariate shifts, which motivates our work.

### F.2  CLIP IN FL: FROM EFFICIENT FINE-TUNING TO DOMAIN GENERALIZATION

The primary barrier to deploying VLMs in FL is their immense computational and communication overhead. Consequently, mainstream research has converged on parameter-efficient fine-tuning (PEFT) (Pan et al., 2024; Saha et al., 2025a; Chen et al., 2024). This strategy freezes the large VLM backbone and exclusively trains lightweight, client-side modules. Approaches range from federated prompt learning (Guo et al., 2023a; Yang et al., 2023), where clients collaboratively tune prompts, to designing specialized adapters that help the global model adapt to local data with minimal communication (Wu et al., 2025; Lu et al., 2023; Shi et al., 2024).

Although PEFT-based methods mitigate the efficiency bottleneck and offer some improvement in generalization, they fail to address the fundamental problem: efficiently trained models tend to learn "spurious features" tethered to specific domains, resulting in poor performance on unseen domains (Varma et al., 2024b). While recent research attempts to guide models toward invariant features (Zhang et al., 2024b; Guo et al., 2025) using techniques like test-time prompt optimization (Ma et al., 2025) and causal learning (Chen et al., 2023; Zhang et al., 2025), these methods share a fundamental flaw from a causal perspective. They erroneously assume that invariant and domain-specific features are independent, thus ignoring the profound causal connections between them. In contrast,

our work is the first to introduce a more complete causal inference framework to federated VLMs. We advocate for modeling the deep causal structure behind the features, positing a core hypothesis: a pure, domain-invariant "concept" acts as the common root cause for the observed invariant features, variant features, and the final label. By performing inference on this causal graph, our model aims to transition from learning "superficial correlations" to understanding "deep causality," thereby fundamentally enhancing its domain generalization.

