# OpenReview forum: "CauFed-CLIP: Causal Federated Vision-Language Models for Domain Generalization"
_ICLR.cc/2026/Conference — ICLR 2026 Conference Withdrawn Submission_

### Official Review · Reviewer_3peE · 2025-10-26

**Soundness:** 2
**Presentation:** 2
**Contribution:** 2
**Rating:** 2
**Confidence:** 5

**Summary:**

The paper proposes CauFed-CLIP, a federated learning framework that adapts a large vision-language model (e.g., CLIP) to heterogeneous clients without sharing raw data. The method introduces a causal factorization of features into invariant, task-relevant information and domain-specific nuisance factors, and trains lightweight prompt/adapter modules on each client while keeping the backbone frozen. Experiments on several image classification benchmarks under domain shift report improvements over prior FL and prompt-based baselines.

**Strengths:**

1. The paper tackles an important and realistic problem: federated learning under strong domain shift, where raw data cannot be shared across clients.

2. The proposed CauFed-CLIP framework is conceptually interesting, it freezes a large vision-language backbone and only trains lightweight prompt / adapter modules, which is attractive for efficiency and potentially for privacy.

**Weaknesses:**

1. The paper defines the federated learning objective as minimizing the loss on each client's “test set.” This is not standard practice: in FL we usually optimize empirical risk on training data and then evaluate on validation/test data. Please clarify whether “test set” here is a typo (and should have been “train” or “local data”), or whether the method truly optimizes on test data. As written, it sounds like you are training on test labels, which is problematic and conceptually inconsistent with FL.

2. The paper describes the edge from the concept variable c to the style/domain variable z as a “weak causal link,” and even introduce a custom notation for it. This notation is non-standard, and “weak causal link” is never defined in terms of parameters or effect size. Please formalize what “weak” means (e.g., a small coefficient in a structural equation, or a regularization that penalizes dependence), and ideally ablate that edge in experiments. Otherwise, it remains a narrative device rather than a verifiable assumption.

3. The method relies on separating invariant features s and domain-specific features z, with guidance from global and local prompts. However, the actual training signal for this separation seems to come from a symmetric contrastive loss over batch instances, where identities in the batch act as pseudo-labels. That is much closer to instance discrimination than to semantic disentanglement. It is not yet clear why this guarantees that s captures causal/semantic content and z captures spurious/domain content instead of just splitting arbitrary directions in feature space. This separation needs clearer justification or empirical evidence.

4. The paper reports results on datasets like PACS, OfficeHome, CIFAR-C, and a medical brain tumor dataset, but the setup is sometimes underspecified. For example, “ModernOffice-31” is mentioned without a clear definition of its source, splits, and licensing status, and the PACS table mixes columns for clients and for domains in a way that is hard to interpret. The paper should clearly explain how clients are formed, how domains are assigned, and which columns correspond to which evaluation scenarios. Without that, it is difficult to verify whether the federated scenario is realistic.

5. The baselines need to be matched more explicitly. Are all methods using the same frozen CLIP backbone? Do they have comparable numbers of trainable parameters, the same prompt length, the same learning rate schedule, and the same communication budget? Since your contribution emphasizes both accuracy and efficiency, you should report the communication cost per round, total rounds to convergence, FLOPs per client per round, and memory footprint for your method and for each baseline. Right now, the claim of efficiency is not quantitatively supported.

6. Some symbols and names are inconsistent across the paper. For example, “FedAvg” vs. “FedAVG,” “CauFed-CLIP” vs. “CAUFED-CLIP,” and “FAA-CLIP” vs. “FAACLIP.” Probability distributions are sometimes written with uppercase P(⋅) and sometimes lowercase p(⋅) without explanation. These inconsistencies can confuse readers and should be harmonized.

7. Some of the reported performance gains are relatively modest, and in some figures it is unclear whether they are statistically significant. Without error bars or significance tests, it is hard to tell if the observed improvements are due to the proposed causal factorization, or if they are simply due to adding more trainable parameters or tuning prompts more aggressively. Adding standard deviation/error bars and stating the number of runs would increase the credibility of the empirical claims.

**Questions:**

1. In Eq. (1), you define the training objective using each client's “test set.” Do you actually optimize on test data, or is this meant to refer to local training data?

2. The causal argument relies on conditioning on z to identify p(y∣c), but z is itself latent. Under what assumptions is z identifiable from observed data, and can you provide either a theoretical statement or an empirical sanity check on identifiability?

3. What exactly does the decoder reconstruct in the variational objective, raw pixels, CLIP embeddings, or something else? How is the reconstruction term weighted relative to the classification and contrastive losses?

---

### Official Review · Reviewer_Gnzg · 2025-10-27

**Soundness:** 2
**Presentation:** 1
**Contribution:** 2
**Rating:** 4
**Confidence:** 4

**Summary:**

The paper proposes CauFed-CLIP, a causal based federated framework based on CLIP. The authors formalize an SCM and derive an ELBO-style training objective that combines a supervised prediction term with a causal consistency regularizer. The architecture uses shared global prompts and private local prompts as semantic anchors. Experiments on domain generalization report consistent gains over previous work.

**Strengths:**

1. The paper translates its SCM into a tractable objective. The decomposition into reconstruction and KL terms is clean and easy to optimize.
2. The experimental results demonstrate the effectiveness of the proposed method.

**Weaknesses:**

1. The problem definition part is somewhat unclear, as Eq. (1) defines the objective as minimizing each client’s test-set loss, and line 151 incorrectly treats covariate shift as distinct from domain shift, even though it is actually a subtype of domain shift.
2. The logical flow of the paper is confusing. In lines 198–199, the authors state, “We assume our framework’s neural networks (Fig. 2) can sufficiently approximate true data distributions like $p(x|s, z, c)$ and $p(y|c)$,” yet the framework itself has not been introduced at this point. Similarly, in lines 213–214, the authors claim that “$p(y|c)$ is theoretically identifiable, provided that our learning algorithm can effectively infer and disentangle the latent variable $z$ from the observed data $x$,” but the learning algorithm has not been described in detail up to this point.
3. It is unclear how the proposed objective function relates to the causal graph analysis, as the paper mentions Pearl’s back-door adjustment yet does not explain in the main text how the SCM connects to the objective; only after reading the appendix does it become apparent that the SCM is used to factorize the joint distribution.
4. The dashed edge $c \to z$ in the causal graph is confusing, since dashed lines in SCMs typically denote bidirectional confounding arcs, whereas here it is a single directed arrow, so the intended meaning is unclear.
5. In the Prompt-Guiding Mechanism, the authors introduce $v^L$ and $v^G$, yet the objective includes only $v^L$.
6. In Fig. 2, the font is too small, and the arrow labels are confusing; the arrows form cycles, which makes the algorithmic procedure hard to follow.

**Questions:**

See the Weakness above

---

### Official Review · Reviewer_ZH2o · 2025-10-29

**Soundness:** 2
**Presentation:** 2
**Contribution:** 2
**Rating:** 2
**Confidence:** 5

**Summary:**

This work presents CauFed-CLIP, which aims to improve federated learning under domain shift by disentangling “causal” content from spurious style factors and only sharing small prompt-like modules instead of full models. The approach is evaluated on multiple visual datasets and is claimed to generalize better to unseen clients. While the idea is interesting and practically motivated (privacy, efficiency), some parts of the causal formulation and experimental setup are not yet fully clarified.

**Strengths:**

1. The paper is well aligned with current interest in adapting large pretrained vision-language models to federated settings. By reusing a frozen CLIP backbone and only learning small client-side modules, the approach is practically appealing for deployment on resource-limited clients.

2. The method aims to separate causal, task-relevant information from domain-specific nuisance factors, and the reported results on several benchmarks suggest consistent gains over existing FL and prompt-based baselines.

**Weaknesses:**

1. The derivation of the variational objective is not entirely consistent across equations. Some terms assume access to qϕ(c,s,z∣x,y), others use qϕ(c,s,z∣x), and later you incorporate
logqϕ(y∣x) together with a causal consistency regularizer. These conditional assumptions are not clearly reconciled. It would be important to show a step-by-step derivation in the main text (not only in the appendix) that explains which variables are observed during training, which ones are latent, and under what approximations you arrive at the final loss in Eq. (6).

2. Equation (7) effectively factorizes pθ(s,z∣c) into independent terms for s and z given c. This is a strong conditional independence assumption. Earlier in the paper, you motivate s and z as two entangled aspects that are both generated from the same “concept” c, which suggests some coupling. Please explicitly state this independence assumption in the main text and argue why it is reasonable. Right now the paper uses it in the math but does not justify it conceptually or empirically.

3. The paper argues that, by conditioning on z, it is possible to identify p(y∣c) using the backdoor criterion. However, z itself is a latent, model-internal quantity rather than an observed confounder. The argument currently reads as “if we could perfectly infer z, then the causal effect is identifiable,” which is circular. The paper should state under what conditions z is identifiable from data, for example via multi-domain variation, anchors, or structural assumptions. Without those conditions, the causal claim is not yet convincing.

4. The choice of baselines is not fully appropriate. The paper mainly compares against older federated learning methods and methods that are not prompt-based, while it does not include several more relevant approaches discussed in the related work section, especially recent prompt-driven FL/adapter-style methods and works that also incorporate causal reasoning. For a fair evaluation, the comparison should include these closer contemporaries, not mainly early baselines that are less aligned with the proposed setting.

5. The size of Figure 2 is not appropriate. It is rendered too small, and important architectural details (modules, arrows, labels) are difficult to read.

6. The motivation of the paper is not clearly established. The introduction mainly states that domain generalization under federated settings is challenging, but it does not analyze in detail what specific limitations in existing methods the proposed approach is intended to fix. This lack of a concrete problem statement also shows up in the experiments: the evaluation does not explicitly demonstrate how the method addresses a clearly identified failure mode of prior work. As a result, the mechanism by which the method is effective remains unclear, and the overall contribution is difficult to interpret.

**Questions:**

1. How are global prompts and local prompts actually aggregated on the server? Are they averaged (FedAvg-style), distilled, or kept private per client? Please describe the communication protocol and its cost in concrete numbers.

2. Several datasets / tables are described but the split definitions (e.g., “ModernOffice-31,” PACS clients/domains) are not entirely clear. Can you provide precise details on how clients are formed, how domains are assigned, and what columns in the tables correspond to which evaluation setting?

3. Do you have any measurement (e.g., bytes per round, FLOPs per client, or membership inference risk) that supports these claims, or can you include such analysis in the final version?

---

### Official Review · Reviewer_afKD · 2025-10-30

**Soundness:** 2
**Presentation:** 2
**Contribution:** 2
**Rating:** 2
**Confidence:** 4

**Summary:**

The paper proposes CauFed-CLIP, a causal federated VLM that freezes the CLIP backbone and trains a lightweight “causal module” on clients to cut compute/communication while improving cross-domain generalization. It disentangles image features x into domain-invariant s and domain-variant z via global (shared) and local (client) prompts, then infers a latent concept c (quasi-invariant, weakly linked to domain z) and predicts using p(y∣c). A prompt-guided variational inference objective couples supervised prediction with a causal consistency regularizer derived from an ELBO reformulation. Experiments across six benchmarks report consistent gains over FL baselines, especially under domain shift.

**Strengths:**

1. The causal graph motivates blocking the back-door path y←C↔z→x by conditioning on z, justifying prediction via p(y∣c) to reduce spurious domain correlations, which is well aligned with FL heterogeneity.
2. Freezing CLIP and training only a small module per client directly targets compute/communication constraints in FL.
3. The global/local prompts provide disentanglement between domain invariant and domain-specific features.
4. Multiple domain generalization and corruption settings, with visualizations (BACC curves, heatmaps) and tables indicating consistent improvements.

**Weaknesses:**

1. The paper infers a causal structure (concept–domain entanglement and a “weak causal” link  c↔z from dataset bias, but offers no interventions/counterfactuals to justify causality beyond correlation.
2. Backdoor adjustment requires probabilistic conditioning/integration, yet the implementation relies on prompts and contrastive/VI losses. The paper does not bridge how these deterministic modules approximate the required probability-weighted adjustment.
3. This paper tends to oversated its contibution. Specifically, this paper claims a “causal deconfounding framework,”but the empirical gains likely stem from the engineering recipe (frozen CLIP + lightweight client module + dual prompt), not from a validated causal intervention.
4. Cross-domain improvements demonstrate effectiveness but do not verify the causal assumption. Missing are direct tests such as intervening on z fixed, measuring confounding reduction, or prompt-sensitivity analyses.
5. Figures are too small and hard to read.

**Questions:**

1. Beyond an intuitive “concept” latent, does c correspond to any measurable or controllable quantity (e.g., object identity, pathology, scene attribute)?
2. Why optimiza an ELBO, What specific benefits does the ELBO-based VI objective provide over a purely discriminative loss with structured regularizers (e.g., prompt alignment, invariance penalties)? Could it possible to provide some ablation study regarding this part?

---

### Note · Authors · 2025-11-12

I have read and agree with the venue's withdrawal policy on behalf of myself and my co-authors.